# EARTHSCAPE: A MULTIMODAL DATASET FOR SURFICIAL GEOLOGIC MAPPING AND EARTH SURFACE ANALYSIS

## ABSTRACT

Surficial geologic (SG) maps are essential for understanding surface processes and supporting infrastructure planning, but current workflows are labor-intensive and difficult to scale. We introduce EarthScape, an AI-ready multimodal dataset for SG mapping that integrates digital elevation models, aerial imagery, multi-scale terrain features, and hydrologic and infrastructure vector data within a unified, reproducible pipeline. We report baseline benchmarks across single-modality, multi-scale, and multimodal configurations. In our experiments, terrain-derived features provide the most reliable predictive signal, while spectral inputs and raw elevation degrade substantially under cross-region evaluation. Cross-generalization and multimodal fusion remain challenging, underscoring the need for models that capture shape-driven surface processes. EarthScape offers a geographically compact but modality-rich benchmark for multimodal fusion, domain adaptation, and surface-process modeling.

## 1 INTRODUCTION

Surficial geologic (SG) maps depict the spatial distribution of mostly unconsolidated materials on the Earth's surface (Compton, 1985). These maps are essential to address a range of contemporary challenges, such as supporting economic and national security interests in critical mineral resources (Brimhall et al., 2005; Schulz, 2017), informing mitigation and response planning for geologic hazards (Alcántara-Ayala, 2002; Van Westen et al., 2003), and providing a foundation on which to understand climate change (Anderson & Ferree, 2010). SG maps are also relevant to more practical applications like urban land use planning (Dai et al., 2001; Hokanson et al., 2019) and engineering projects (Keaton, 2013). Despite the demonstrable social benefit and scientific merit (Bernknopf, 1993), detailed SG maps cover less than 14% of the United States (U.S. Geological Survey, 2025), and coverage is even more limited globally.

The modern SG mapping workflow relies on manual fieldwork coupled with visual interpretation of remote sensing (RS) imagery (Compton, 1985; Lisle et al., 2011). Because SG maps depend on expert interpretation and annotation, they may reflect local subjectivity, rather than reproducible, global criteria. Moreover, financial costs are prohibitive, with one standard 1:24k-scale map[1] estimated at $123k (Berg, 2025). These limitations highlight the need for scalable, automated approaches.

Advancements in deep learning and the proliferation of RS imagery present an opportunity to transform SG mapping and overcome current limitations. Recent studies have demonstrated the potential of deep learning to identify or segment single class geologic hazards, such as landslides (Prakash et al., 2021; Wang et al., 2021; Liu et al., 2023) and sinkholes (Rafique et al., 2022), and a few have extended these ideas to mapping multiple classes of geologic materials (Behrens et al., 2018; Latifovic et al., 2018; Wang et al., 2021; Liu et al., 2024b). While these works highlight the promise of computer vision (CV), they remain constrained by narrow scope, limited modality integration, and the absence of standardized benchmarks.

---

[1]Map scale refers to cartographic accuracy, rather than raster resolution. At 1:24,000-scale, one map unit represents 24,000 real-world units, and is considered the gold-standard geologic mapping scale.

The challenges of SG mapping align closely with current directions in CV. Multimodal fusion of heterogeneous inputs is required to capture features invisible to any single modality (Baltrušaitis et al., 2018; Steyaert et al., 2023; Li & Wu, 2024). Strong spatial dependencies make it a natural testbed for attention mechanisms and multi-scale architectures (Dosovitskiy, 2020; Niu et al., 2021; Fan et al., 2021; Hassanin et al., 2024; Liu et al., 2024a), while extreme class imbalance and geographic variability mirror open challenges in long-tail learning and domain adaptation (Lin, 2017; Ghosh et al., 2024). Beyond SG mapping, surface morphology is an underutilized signal across domains such as medical imaging where shape descriptors from CT or MRI improve disease prediction (Van Timmeren et al., 2020), autonomous navigation where terrain guides safe decision-making (Meng et al., 2023), and RS where benchmarks often underemphasize topography (Wang et al., 2025).

The rapid progress in CV has been driven by the availability of large-scale, standardized datasets. General-purpose benchmarks like ImageNet (Deng et al., 2009) and COCO (Lin et al., 2014) have catalyzed advances in classification, detection, and segmentation by offering vast repositories of labeled imagery and clear evaluation protocols. However, performance on real-world tasks often plateaus without domain-specific datasets that reflect their unique characteristics, sensing modalities, and physical constraints. In the geospatial domain, datasets have emerged for land cover classification and urban scene analysis (Schmitt et al., 2019; Cordts et al., 2016; Demir et al., 2018; Van Etten et al., 2018; Sumbul et al., 2019), but these are primarily for anthropogenic features and land use. Several geologic datasets have been introduced for hazard mapping, but these focus on discrete events (Ji et al., 2020; Montello et al., 2022; Rege Cambrin & Garza, 2024), leaving a critical gap in geoscience datasets tailored to more realistic conditions with continuous materials.

EarthScape is a multimodal dataset developed for SG mapping, with applicability to other surface-aware geospatial tasks. It integrates publicly available RGB and near-infrared (NIR) imagery, digital elevation models (DEM), DEM-derived terrain features computed at multiple scales, and transportation and hydrological vector data into a unified, co-registered framework. This design reflects key characteristics of SG mapping, including multi-label structure, scale-dependent morphology, and geographic heterogeneity, and provides a benchmark for developing and evaluating multimodal geospatial models. Our contributions are as follows:

- We introduce EarthScape, the first multimodal, multi-scale benchmark dataset designed specifically for SG mapping and surface-aware geospatial analysis.

- We provide a unified, co-registered framework integrating imagery, elevation, multi-scale terrain derivatives, and vector layers, enabling flexible multimodal experimentation.

- We establish reproducible baselines across unimodal, multi-scale, and multimodal configurations, supporting systematic evaluation of fusion strategies, backbone architectures, and cross-domain generalization.

## 2 RELATED WORK

**SG Mapping with Machine Learning:** SG mapping focuses on unconsolidated materials formed by active surface processes, such as weathering, erosion, sediment transport, and deposition (Compton, 1985). These materials are closely tied to landform structure and surface morphology, as terrain shape governs the energy available to drive these processes (Odeh et al., 1991; Schomberg et al., 2005; Brigham & Crider, 2022). Several studies have leveraged this terrain-geologic material relationship using logistic regression, random forests, and support vector machines for classification or segmentation of binary hazards (e.g., landslides, sinkholes) (Kirkwood et al., 2016; Zhu & Pierskalla Jr, 2016; Crawford et al., 2021) or SG maps (Cracknell & Reading, 2014; Johnson & Haneberg, 2025). However, these approaches depend on hand-crafted features, are restricted to small geographic extents, and fail to generalize beyond the training region. More recently, deep learning methods using convolutional neural networks (CNNs) and CNN-Transformer hybrids have been applied to related tasks (Prakash et al., 2021; Ji et al., 2020; Liu et al., 2023; Latifovic et al., 2018; Zhou et al., 2023; Rafique et al., 2022). While these models better capture spatial dependencies critical to geologic interpretation (Bishop et al., 1998; Behrens et al., 2018), they remain site-specific, lack standardized datasets, and rely on limited input modalities.

**Multimodal Learning for Geologic Tasks:** Multimodal learning has become a central paradigm in geospatial CV, where combining diverse data sources, like optical imagery, SAR, and DEMs,

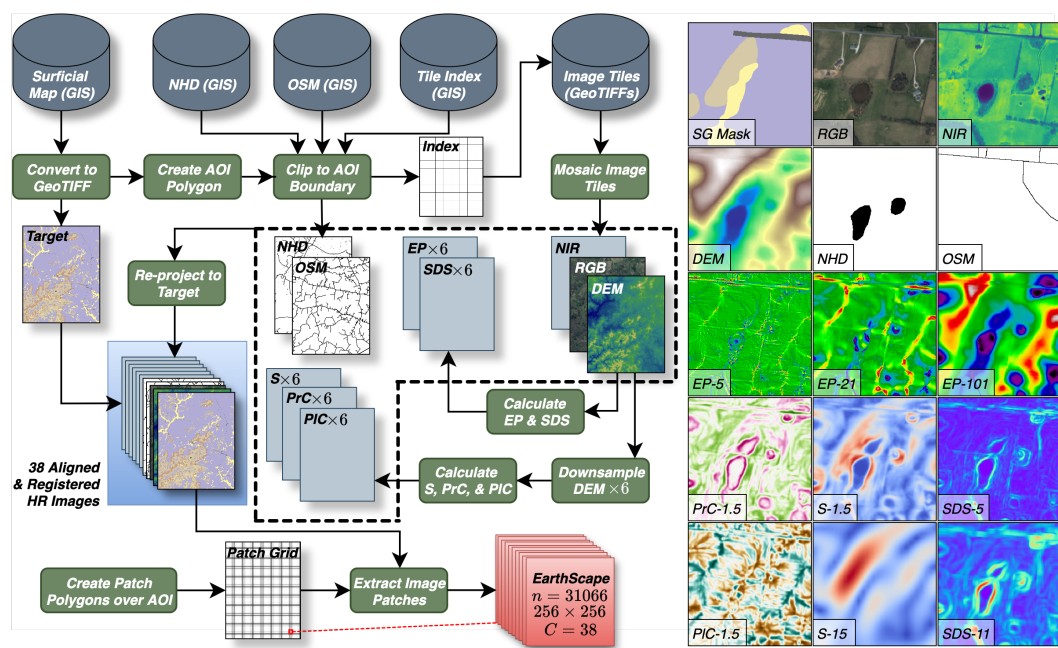

Figure 1: EarthScape data processing pipeline (left) and selected modalities from a single $256 \times 256$ patch (right). The SG map is rasterized and used to define the area of interest (AOI), from which all predictive features (DEM, RGB+NIR imagery, NHD hydrology, and OSM infrastructure) are clipped and aligned. Terrain derivatives are then computed from the DEM at multiple spatial scales. A regular grid is applied to extract 38 co-registered channels per patch.

can enhance model robustness through learned complementary information (Astruc et al., 2024; Bi et al., 2022; Jain et al., 2022; Han et al., 2024). In geological applications, this has often manifested by fusing overhead RGB imagery with DEMs with early- or mid-level strategies (Prakash et al., 2021; Ji et al., 2020; Liu et al., 2023; Latifovic et al., 2018; Zhou et al., 2023; Rafique et al., 2022). Although effective for some situations, these approaches tend to overfit to absolute elevation or local appearance and fail to generalize to new regions. Other modalities have also been tested, including elevation contours (Zhou et al., 2023), geochemical maps (Latifovic et al., 2018; Wang et al., 2021), and aeromagnetic imagery (Liu et al., 2024b), but these resources lack standardized availability.

**RS and Geologic Datasets:** RS benchmarks like SpaceNet (Van Etten et al., 2018), xView (Lam et al., 2018), and the Functional Map of the World (Christie et al., 2018) provide high-resolution satellite imagery annotated for object detection and scene classification in urban environments. These datasets are optimized for anthropogenic features such as roads, buildings, and vehicles, and are widely used for infrastructure monitoring and disaster response. Other RS datasets, including BigEarthNet (Sumbul et al., 2019), DeepGlobe (Demir et al., 2018), and SEN12MS (Schmitt et al., 2019), support land cover classification and segmentation using multispectral or synthetic aperture radar (SAR) imagery. However, these datasets target coarse semantic categories such as vegetation or developed areas and lack representations of Earth's surface necessary to understand SG processes.

Several geoscience-specific datasets have been introduced for geologic hazards, including MMFlood for flood delineation (Montello et al., 2022), QuakeSet for earthquake event detection (Rege Cambrin & Garza, 2024), and landslide detection datasets leveraging overhead imagery and DEMs (Ji et al., 2020; Liu et al., 2023; Zhou et al., 2023). While valuable for their respective domains, these resources are narrowly scoped to discrete hazards or events, often limited to small geographic areas, and rely on shallow modality combinations. Prior machine learning work on SG mapping similarly relies on small, locally assembled datasets that are not publicly released or standardized (Kirkwood et al., 2016; Zhu & Pierskalla Jr, 2016; Latifovic et al., 2018; Crawford et al., 2021; Johnson & Haneberg, 2025), making systematic comparison and cross-region evaluation impossible. None of these resources supports continuous SG mapping.

## 3 EARTHSCAPE DATASET

### 3.1 DATA SOURCES AND COMPOSITION

**Surficial Geologic Maps:** The EarthScape dataset currently includes eight high-resolution (1:24,000-scale) SG maps covering two areas in the central United States (Buchanan et al., 2023; Massey et al., 2023; Swallom et al., 2023; Massey et al., 2024; Hodelka et al., 2024; Swallom et al., 2024; Bottoms et al., 2021; Massey et al., 2021). Each map is delivered as a vector polygon dataset in ESRI geodatabase format and are rasterized during preprocessing to produce the targets used throughout the benchmark. EarthScape includes seven SG units that form a mutually exclusive representation of the surficial cover in each area. These units correspond to five surface-process environments: fluvial deposits (*Qal, alluvium*; *Qat, terrace deposits*), debris-flow deposits (*Qaf, alluvial fans*), hillslope materials (*Qc, colluvium*; *Qca, colluvial aprons*), in-situ weathering products (*Qr, residuum*), and anthropogenic modification (*af1, artificial fill*). Although EarthScape v1.0 is geographically limited, the mapped environments and surface processes it captures are widespread in temperate, non-glaciated landscapes worldwide. As a result, the SG units in Earth-Scape provide a representative set of classes for evaluating multimodal models designed to generalize across similar geomorphic settings. See Appendices B.1 and B.2 for additional information.

**Aerial imagery and DEM:** EarthScape includes aerial RGB+NIR imagery and LiDAR-derived DEMs (Commonwealth of Kentucky, 2024), which constitute the core RS modalities in the dataset. The aerial imagery has a ground sampling distance (GSD) of 0.15 m ($\approx$ 6 in) and provides measurements of surface appearance: RGB channels capture visible-wavelength variation related to land cover and human modification: NIR band emphasizes vegetation moisture and canopy structure. The DEM is produced from airborne LiDAR with 1.52 m GSD ($\approx$ 5 ft) resolution and provides raw elevation and surface morphology information. Variations in topography, local relief, and slope often align with boundaries between SG materials, making DEM data an intuitive modality for SG mapping tasks. Both datasets are publicly accessible as GeoTIFF tiles and are co-registered during preprocessing to ensure consistent spatial alignment with all other EarthScape modalities.

**Terrain Features:** EarthScape includes five DEM-derived terrain features widely used in geomorphometry (Florinsky, 2016), each quantifying a distinct aspect of surface geometry. *Slope (S)* describes local surface steepness; *profile curvature (PrC)* and *planform curvature (PlC)* capture surface curvature parallel and perpendicular to the direction of maximum slope; *elevation percentile (EP)* measures relative elevation; *standard deviation of slope (SDS)* characterizes local surface roughness. See Appendix B.3 for more information.

**Hydrography and Infrastructure:** EarthScape includes vector data for surface hydrography and human infrastructure. Hydrographic features consist of stream centerlines and waterbody polygons from the U.S. Geological Survey's National Hydrography Dataset (NHD) (U.S. Geological Survey, 2024), and infrastructure features include road and railway centerlines from OpenStreetMap (OSM) (OpenStreetMap contributors, 2024). These layers supply contextual information about drainage networks and built environments that complements the imagery and terrain features.

### 3.2 DATA PROCESSING PIPELINE

**Targets:** Each SG map was provided as a vector geodatabase, and the relevant polygons exported to a non-proprietary GeoJSON format (Fig. 1). The polygons were checked for valid geometry and their topology was validated to ensure complete coverage, preventing gaps or inconsistencies that could produce missing or incorrect labels during rasterization. All SG units were then mapped to a standardized set of ordinal class values shared across the entire EarthScape dataset. The vector data were reprojected to the DEM coordinate reference system and rasterized to a common 1.52 m GSD grid (Fig. 1). The DEM was used as the target grid because it served as the original basemap for the mapping and provides a uniform reference for aligning all other modalities.

**Raw Features:** A tile index defining the footprints of the RGB+NIR imagery and DEM tiles was obtained, and all tiles intersecting the AOI were downloaded (Fig. 1). The aerial RGB+NIR and DEM GeoTIFF tiles were reprojected and merged into single raster mosaics at a common 1.52 m GSD resolution (Fig. 1). Vector hydrography and infrastructure datasets were also acquired and clipped to the AOI (Fig. 1). NHD hydrographic and OSM infrastructure features were then rasterized into two binary GeoTIFF layers aligned to the same 1.52 m GSD grid (Fig. 1).

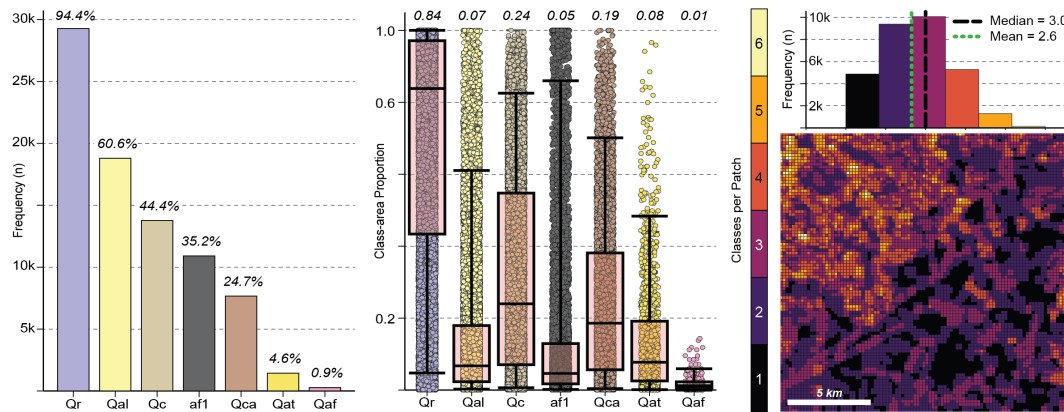

Figure 2: EarthScape label distribution summaries. Left: Global class frequencies ordered by descending prevalence; relative frequencies shown above each bar. Center: Patch-level class-area distributions shown as class-area proportion values and boxplots (interquartile range with whiskers to the 5th–95th percentiles); median values displayed at top. Right: Histogram (top) and an example area map (bottom) each symbolized by its per-patch class count.

Table 1: Label statistics and imbalance metrics for EarthScape, including global frequency, class-area proportion (mean and SD), majority area rate (MAR), effective number of samples (ENS) (Cui et al., 2019), and the imbalance ratio per label (IRLbl) (Charte et al., 2013).

| Class | Frequency ($n$) | Frequency (%) | Mean Class-area | SD Class-area | MAR | ENS | IRLbl |
|-------|-----------------|---------------|-----------------|---------------|-----|-----|-------|
| Qr    | 29271 | 94.4 | 0.651 | 0.358 | 0.702 | 9464.6 | 1.0 |
| Qal   | 18801 | 60.6 | 0.089 | 0.168 | 0.058 | 8474.4 | 1.6 |
| Qc    | 13768 | 44.4 | 0.142 | 0.242 | 0.148 | 7476.3 | 2.1 |
| af1   | 10910 | 35.2 | 0.051 | 0.161 | 0.035 | 6641.4 | 2.7 |
| Qca   | 7669  | 24.7 | 0.061 | 0.154 | 0.054 | 5355.7 | 3.8 |
| Qat   | 1435  | 4.6  | 0.006 | 0.045 | 0.004 | 1336.9 | 20.4 |
| Qaf   | 270   | 0.9  | 0.000 | 0.003 | 0.000 | 266.4  | 108.4 |

**Engineered Features:** Terrain features were calculated at multiple spatial scales in order to capture hierarchical surface structure (Fig. 1). The native DEM (1.52 m GSD) was downsampled to five additional resolutions (3.05, 6.10, 15.24, 30.48, 60.96 m GSD) following a roughly logarithmic progression commonly used in geomorphometry (Fig. 1). S, PrC, and PlC were computed on each DEM using 5×5 neighborhood kernels, upsampled back to 1.52 m GSD (Fig. 1), and smoothed with a Gaussian filter to reduce interpolation artifacts. EP and SDS were computed directly on the native-resolution DEM as neighborhood statistics using kernels of 5×5, 11×11, 21×21, 51×51, 101×101, and 201×201 pixels (Fig. 1). Kernel sizes were chosen so that their effective spatial footprint matches the approximate resolutions used for S, PrC, and PlC, ensuring comparable multi-scale representations across modalities. Additional details are provided in Appendix B.3.

**Spatial Alignment and Registration:** The rasterized SG map served as the reference grid for the entire dataset. Each rasterized feature was reprojected to a common coordinate reference system to ensure identical spatial resolution, grid origin, and geographic extent (Fig. 1). After reprojection, all images were validated to confirm matching bounding coordinates and pixel dimensions, guaranteeing full spatial alignment across modalities.

**Patches:** Vector polygons were constructed in a systematic grid to cover each SG map AOI (Fig. 1). Each patch is 256×256 pixels (390×390 m), overlaps adjacent cells by 50%, and is constrained to lie completely within the AOI. The 256×256 patch size was selected so that identifying geomorphic features mapped at 1:24,000-scale typically fall within an individual patch, while the overlapping design enables users to construct larger effective context windows if needed. Each patch received a unique ID and was used to extract all 38 channels from the aligned modalities (Figs. 1, 6–7). For each patch, area proportions were computed from the SG mask to summarize class presence.

## 3.3 Dataset Properties and Statistics

**Overview and Structure:** EarthScape currently comprises 31,018 georeferenced patches from two geographic regions. Each patch is 256×256 pixels with 50% overlap and contains 38 co-registered channels, including the mask, RGB+NIR imagery, DEM, multi-scale terrain derivatives, and binary hydrography and infrastructure layers. EarthScape includes seven SG units. Each patch includes the pixel-level SG mask and proportional class-area summaries, enabling multilabel classification, semantic segmentation, regression, and multitask configurations. See Appendix A.3 for more details.

**Class Distribution and Imbalance:** EarthScape exhibits a pronounced long-tailed distribution across its seven SG units (Table 1; Fig. 2). Qr appears in 94.4% of patches, whereas the rarest units occur in only 4.6% (Qat) and 0.9% (Qaf) of patches. Effective number of samples ranges from 9,464 (Qr) to 266 (Qaf), and the imbalance ratio per label spans more than two orders of magnitude (1.0-108.4), reflecting strong label-level complexity driven by frequency skew. Beyond global frequencies, EarthScape exhibits marked intra-patch complexity. Mean and standard-deviation class-area proportions show that most patches contain multiple SG units with uneven contributions, and the majority-area rate indicates that Qr dominates more than 70% of patches while rare units almost never occupy the largest fraction. Patch-level class counts vary widely across the regions, reflecting strong geospatial complexity in how classes co-occur and mix spatially.

**Domain Shift:** EarthScape spans two disjoint regions in Kentucky, USA, consisting of 23,566 patches from Warren County and 7,452 patches from Hardin County, separated by nearly 75 km. This structure provides a natural geographic partition for analyzing cross-region variation. We compute maximum mean discrepancy (MMD) to quantify distributional differences between patch-level feature summaries (P10, P25, P50, P75, P90) of selected input modalities from each region (Gretton et al., 2012). We observe measurable domain shift (Table 7), including MMD values of 0.365 for RGB, 0.832 for DEM, and 0.164 for a multi-scale terrain stack (EP+S+SDS). Although both regions share the same label set, their input feature distributions differ, reflecting geographic variation and providing a clean, geographically partitioned setting for studying domain shift in multimodal geospatial learning. See Appendix C.4 for additional details.

# 4 Experiments

## 4.1 Methods

**Task Definition:** We formulate SG mapping as a multilabel classification task over multimodal geospatial inputs. Each input sample corresponds to a $256 \times 256$ image patch with co-registered modalities and a label vector indicating the presence or absence of each of the SG units. Let $\mathcal{D} = (x_i, y_i)_{i=1}^{N}$ denote the dataset, where each $x_i = m_1, m_2, \ldots, m_n$ is a collection of $n$ modality-specific input tensors (e.g., DEM, EP, PlC, etc.) and each modality $m_i$ can have multiple scaled images that we consider as channels $C_i$. The $y_i \in 0, 1^K$ is a binary label vector over $K = 7$ classes, where a class is marked positive if any part of its mask intersects the patch (i.e., even a single pixel), without applying a proportional threshold. The model learns a mapping $f : X \to [0, 1]^K$ to predict per-class probabilities, enabling multi-class label assignment for each patch. This formulation allows us to systematically evaluate how different modality combinations contribute to SG feature recognition and serves as a tractable benchmark for future tasks.

**Surficial Geologic Mapping Network (SGMap-Net):** We introduce SGMap-Net as a lightweight model designed to effectively integrate the complementary information across modalities and serve as a transparent and interpretable baseline. Its simplicity allows us to isolate the contributions of modality and fusion strategy without architectural confounds, while ensuring that results are reproducible and easily extendable. Figure 3 illustrates the architecture of SGMap-Net, which consists of three key components: a standardization module, a feature extractor, and a classification head. As part of our early fusion strategy, we first stack all channels of each modality $m_i$ and then apply a $1 \times 1$ convolution followed by batch normalization and ReLU activation to standardize the input to a common channel dimension $C = 3$. This ensures compatibility with a shared encoder, while preserving modality-specific spatial patterns through independent convolutions.

$$\hat{m_i} = \text{ReLU}(\text{BN}(\text{Conv}1 \times 1(m_i))). \tag{1}$$

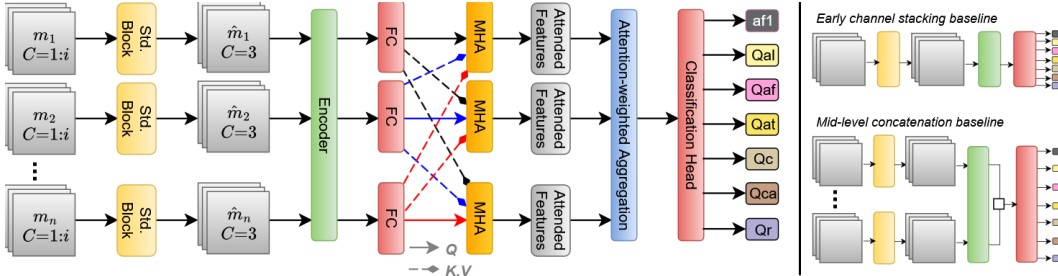

Figure 3: SGMap-Net and fusion baselines. Left: SGMap-Net accepts any number of modalities with arbitrary channels, standardizes each to a 3-channel representation, and encodes them with a shared encoder. Modality features are projected into a latent space for multi-head attention (MHA) and fused via attention-weighted aggregation before classification. Right: Fusion baselines used in experiments, including early channel stacking (top) and mid-level feature concatenation (bottom).

Each standardized modality $\hat{m}_i$ is passed through a shared encoder to extract feature maps $f_{m_i} = \text{Encoder}(\hat{m}_i)$; we experiment with ResNeXt-50 (Xie et al., 2017) and Vision Transformer (ViT-B/16) (Dosovitskiy, 2020) backbones initialized with ImageNet-pretrained weights. Next, each feature vector $f_{m_i}$ is projected into a common latent space of dimension $d$ using a fully connected layer and augmented with a learnable modality embedding $e_i$ to get the final representations $z_i = f_{m_i} + e_i$. Then we apply modality-specific multi-head attention (MHA) (Vaswani et al., 2017) mechanisms to enable intermediate fusion across modalities. For each modality $m_i$, attention is computed using $z_i$ as the query $(Q)$, and the embeddings from all other modalities as keys $(K)$ and values $(V)$.

$$a_i = \text{MHA}(Q = z_i, K = [z_j]_{j \neq i}, V = [z_j]_{j \neq i}). \tag{2}$$

Next, we perform attention-weighted aggregation over the set of modality-specific attention outputs $a$. We begin by concatenating all outputs $A = [a_i]$. To determine the relative importance of each modality, we apply a learnable linear projection $v_i$ followed by a Softmax operation to obtain attention weights $w = \text{Softmax}(v^T A)$. The final fused representation is then computed using these weights, $z_{fused} = \sum_{i=1}^{N} w_i a_i$. This attention-weighted aggregation adaptively emphasizes the most informative modalities for each sample. The fused embedding $z_{\text{fused}}$ is then passed through a classification head consisting of two fully connected layers to predict the geologic class logits $\hat{y}$. In addition to our proposed attention-based fusion strategy, two alternative approaches are evaluated: (1) we stack selected channels from different modalities, extract a joint representation using the encoder, and feed it into the classification head; (2) we concatenate modality embeddings from the encoder and pass them directly to the classification head. These variants serve as comparative baselines to assess the impact of modality-aware attention in our fusion framework.

**Data Splits and Selection:** We define training, validation, and test splits using the Warren County subset, all selected using a fixed random seed. We randomly sampled 1,536 patches for the in-domain test set, then 768 non-intersecting patches for validation, and the remaining 8,416 non-intersecting patches formed the training set (Table 5; Fig. 8a). A cross-domain test set of 1,536 patches was sampled from Hardin County (Table 5; Fig. 8b). All splits exhibit similar class distributions (Fig. 9). This benchmark split preserves spatial independence, reflects standard dataset proportions, and enables clear comparison between in-domain and cross-domain performance.

**Training Procedure:** Each modality was normalized using channel-specific means and standard deviations computed from the training set. Data augmentation included random flips and $90°$ rotations that preserve surface structure, while avoiding potential label mismatch from arbitrary-angle rotations. To address class imbalance, we used focal loss (Lin, 2017) with $\alpha = 0.25$ and $\gamma = 2.0$; oversampling was tested, but reduced performance. Models were trained for 15 epochs with Adam (learning rate 0.001, batch size 16), and the checkpoint with the lowest validation loss was used for evaluation. Label-wise decision thresholds were tuned on the validation set and applied to both test sets. Performance is reported using per-class and macro-averaged precision, recall, F1, AP, and AUC. See Appendices C.2 and C.3 for additional hardware, compute, and focal loss details.

## 4.2 RESULTS AND DISCUSSION

**Modality Performance:** Across single-modality experiments, terrain features provide the strongest overall performance (Tables 2, 8–10; Fig. 11). EP achieves the highest in-domain F1 (0.651), followed by S (0.647), both outperforming RGB (0.599) and DEM (0.632). Under cross-region evaluation, EP and RGB exhibit the largest degradations (0.291, 0.267), whereas S shows a much smaller drop (0.049). DEM shows moderate degradation (0.105), but is less robust than its terrain derivatives. Multi-scale EP and S do not exceed their best single-scale versions, but they improve cross-region performance (0.068, 0.043) (Tables 2, 11–13; Fig. 11). The strongest configuration is a multi-scale, multimodal input of EP+S+SDS, which has the highest in-domain (0.657) and cross-domain (0.598) F1 scores across all experiments (Tables 2, 14–16; Fig. 11). Adding RGB and DEM to this configuration reduces performance, indicating that raw appearance and elevation is less invariant across regions and can dilute more stable, shape-based information from the terrain derivatives. Overall, terrain features provide the most discriminative and robust representation, and their complementary geometric cues combine more effectively than raw appearance or elevation.

**Cross-domain Performance:** Cross-region performance exhibits qualitative correspondence with the patch-level distributional differences measured by MMD (Tables 2 and 7; see also Tables 8–16 and Fig. 11). RGB shows moderate shift (0.365) and the largest F1 degradation (0.267), reflecting sensitivity to location-specific appearance. DEM exhibits the highest shift (0.832), but generalizes better than RGB, suggesting that raw elevation provides some transferable signal. EP performs well in-domain, but shows moderate shift (0.244) and a large F1 drop (0.271), consistent with region-specific variation in local relief. S and SDS have the lowest shifts (0.097, 0.078) and exhibit strong transfer performance (0.070, 0.060), indicating that these shape-based features provide more region-invariant cues. Multi-scale S shows slightly higher shift (0.155), but improves cross-region robustness (0.637). Multi-scale EP+S+SDS shows similar shift (0.164) and achieves the strongest overall transfer (0.059). While MMD does not directly predict performance, modalities with smaller input distribution differences tend to transfer more reliably across regions.

**Per Class Behavior:** Class-wise AUC varies substantially across units and cannot be explained by frequency alone (Tables 1, 17–18; Figs. 2, 12). Qr appears in 94.4% of patches and achieves strong in- and cross-domain AUC (0.933/0.905), yet Qc shows even higher separability (0.975/0.982) while occurring in only 44.4% of patches. Conversely, Qal is the second most common unit (60.6%), but yields the lowest AUC (0.840/0.760). Rare units are surprisingly separable, with Qat (4.6%) and Qaf (0.9%) achieving competitive AUC values (0.903/0.847 and 0.926/0.964), indicating that distinct spatial expression can offset low prevalence. In our benchmarks, no single modality or scale maximizes AUC across all units. In-domain separability is often highest with multi-scale inputs, while cross-domain robustness tends to be strongest with single-scale features that exhibit lower distributional shift. Overall, per-class performance is shaped by the interaction of frequency, patch-level mixing, spatial footprint, and scale-dependent expression of each class.

**Fusion and Backbone Effects:** Across fusion strategies, early channel stacking consistently yields the strongest performance, followed by mid-level concatenation, and then attention-based fusion (Tables 2, 11–18; Figs. 11–12). Backbone differences are more modest but systematic. ResNeXt-50 and ViT-B/16 achieve their highest scores with stacking, while ViT-B/16 tends to outperform ResNeXt-50 when attention-based fusion is used. Class-wise trends show similar structure. With single-modality inputs, ResNeXt-50 attains higher separability (AUC) for af1, Qal, Qaf, and Qat, whereas ViT-B/16 performs better on Qc, Qca, and Qr. Multi-scale and multimodal configurations improve class-wise performance for both encoders, but largely preserve these relative patterns, suggesting that the two backbones emphasize different aspects of the same inputs. From a geologic standpoint, the SG units where each backbone performs best share similar surface expressions. The units where ResNeXt-50 generalizes well tend to be smaller in spatial extent, lower-relief, and more linear in form, whereas the units where ViT-B/16 performs best exhibit broader, regionally extensive geomorphic patterns. Together, these results show that fusion strategy drives overall robustness, while backbone choice primarily shapes how performance gains distribute across individual classes.

**Comparison with Baselines:** We compare SGMap-Net to several recent multimodal RS foundation models, including DOFA (Xiong et al., 2024), Panopticon-FM (Waldmann et al., 2025), SatMAE (Cong et al., 2022), and SatMAE++ (Noman et al., 2024) (Table 2). SGMap-Net achieves the strongest overall performance. Its multimodal, terrain-only EP+S+SDS configuration attains the highest in-domain F1 (0.657), the best cross-domain F1 (0.598), and the smallest performance drop

Table 2: Macro-F1 and AUC for in-domain (ID), cross-domain (CD), and cross-region degradation ($\Delta$) across selected single-modality, multi-scale, and multimodal experiments. The upper block reports SGMap-Net results and the lower block reports performance of existing RS foundation models. Modality abbreviations follow Section 3.1. Subscripts indicate either the DEM resolution used to compute S, PrC, or PlC (e.g., $S_{1.5}$ from the 1.5 m DEM), the kernel size for EP or SDS (e.g., $EP_{51}$ uses a 51×51 kernel), or multi-scale stacks of all resolutions (e.g., $S_{ms}$). The best and second-best scores in each column are shown in **bold** and underlined, respectively.

| Model | Modality | Fusion | F1 | | | AUC | | |
| --- | --- | --- | --- | --- | --- | --- | --- | --- |
| | | | ID | CD | $\Delta$ | ID | CD | $\Delta$ |
| SGMap-Net (ResNeXt) | RGB | - | 0.599 | 0.394 | 0.205 | 0.815 | 0.557 | 0.258 |
| SGMap-Net (ViT) | RGB | - | 0.579 | 0.332 | 0.267 | 0.793 | 0.526 | 0.267 |
| SGMap-Net (ResNeXt) | DEM | - | 0.632 | 0.527 | 0.105 | 0.883 | 0.730 | 0.153 |
| SGMap-Net (ViT) | DEM | - | 0.618 | 0.512 | 0.237 | 0.857 | 0.620 | 0.237 |
| SGMap-Net (ResNeXt) | $EP_{51}$ | - | 0.651 | 0.380 | 0.271 | 0.876 | 0.663 | 0.213 |
| SGMap-Net (ViT) | $EP_{51}$ | - | 0.604 | 0.489 | 0.078 | 0.835 | 0.757 | 0.078 |
| SGMap-Net (ResNeXt) | $S_{1.5}$ | - | 0.645 | 0.575 | 0.070 | 0.876 | **0.808** | 0.068 |
| SGMap-Net (ViT) | $S_{1.5}$ | - | 0.623 | 0.552 | 0.093 | 0.855 | 0.762 | 0.093 |
| SGMap-Net (ResNeXt) | $S_{ms}$ | Attention | 0.494 | 0.426 | 0.068 | 0.500 | 0.500 | **0.000** |
| SGMap-Net (ViT) | $S_{ms}$ | Attention | 0.557 | 0.519 | 0.038 | 0.615 | 0.594 | 0.021 |
| SGMap-Net (ResNeXt) | $S_{ms}$ | Stacking | 0.637 | 0.594 | 0.043 | 0.864 | 0.804 | 0.061 |
| SGMap-Net (ViT) | $S_{ms}$ | Stacking | 0.593 | 0.533 | 0.061 | 0.798 | 0.705 | 0.093 |
| SGMap-Net (ResNeXt) | $EP_{ms}+S_{ms}+SDS_{ms}$ | Attention | 0.561 | 0.532 | **0.029** | 0.677 | 0.707 | -0.030 |
| SGMap-Net (ViT) | $EP_{ms}+S_{ms}+SDS_{ms}$ | Attention | 0.567 | 0.538 | **0.029** | 0.776 | 0.678 | 0.098 |
| SGMap-Net (ResNeXt) | $EP_{ms}+S_{ms}+SDS_{ms}$ | Stacking | **0.657** | **0.598** | 0.059 | 0.882 | 0.806 | 0.076 |
| SGMap-Net (ViT) | $EP_{ms}+S_{ms}+SDS_{ms}$ | Stacking | 0.621 | 0.569 | 0.053 | 0.860 | 0.774 | 0.086 |
| DOFA | RGB+NIR | - | 0.597 | 0.533 | 0.064 | 0.652 | 0.623 | 0.029 |
| Panopticon-FM | RGB+NIR | - | 0.570 | 0.313 | 0.257 | 0.635 | 0.533 | 0.102 |
| SatMAE | $RGB+DEM+EP_{ms}+S_{ms}+SDS_{ms}$ | - | 0.614 | 0.427 | 0.187 | 0.864 | 0.735 | 0.129 |
| SatMAE++ | $RGB+DEM+EP_{ms}+S_{ms}+SDS_{ms}$ | - | 0.656 | 0.454 | 0.202 | **0.904** | 0.762 | 0.142 |

across regions (0.059). Pretrained models show weaker transfer when used with their native spectral inputs. DOFA reaches an in-domain F1 of 0.597 and a cross-domain score of 0.533, but with a competitive drop (0.064), while Panopticon-FM exhibits severe cross-domain collapse (0.257). To enable a more comparable evaluation, we extended SatMAE and SatMAE++ to accept terrain channels. Although SatMAE++ achieves a strong in-domain F1 (0.656), its cross-domain performance degrades sharply (drop of 0.202). These results indicate that pretrained spectral representations exhibit substantial geographic sensitivity on this task, whereas terrain derivatives provide far more stable cues under region shift. SGMap-Net's use of multi-scale, shape-based geomorphic features therefore yields significantly stronger and more consistent performance, despite its simplicity.

## 5 Challenges and Limitations

**Geographic Scope:** EarthScape v1.0 is sampled from two regions in the central United States. Although compact, this spatial footprint keeps mapping standards, labeling conventions, and sensing modalities consistent, simplifying interpretation and enabling clean, repeatable experiments. Both regions differ enough to induce a measurable domain shift in our benchmarks. Future releases of EarthScape will expand geographic coverage.

**Modality Depth:** EarthScape trades geographic breadth for modality depth. Although the spatial extent is modest, each patch provides 38 co-registered channels of imagery, elevation, multi-scale terrain derivatives, and vector features. This depth emphasizes surface-aware multimodal learning and offers flexibility in inputs and architectures, but also increases dimensionality and complexity.

**Class Imbalance:** EarthScape contains seven SG units with long-tailed distributions. Many units occupy only small portions of a patch, patches often have multiple units, and class presence varies across space. This structure reflects the true distribution of SG materials, but requires models to handle class imbalance, intra-patch complexity, co-occurrence patterns, and spatial heterogeneity.

**Domain Shift:** SG units are governed by surface processes that recur globally, but input RS modalities vary geographically. Models that rely heavily on location-specific cues, such as RGB appearance or raw elevation, exhibit substantial cross-region degradation, whereas terrain-derived features

transfer more reliably. EarthScape's cross-region design makes this explicit and provides a controlled setting for studying domain shift in multimodal geospatial learning.

**Multi-scale and Multimodal Complexity:** SG units are expressed by surface processes spanning a range of spatial scales. EarthScape includes terrain derivatives at six resolutions so that models can learn both fine-scale patterns and broader positional context. Our results indicate that no single scale optimizes performance for all classes, and multi-scale combinations generalize better than single-scale variants. Modality follows the same pattern, and multi-scale, multimodal configurations consistently outperform. This demonstrates the necessity of multi-scale, multimodal fusion and scale-aware architectures, but also increases feature dimensionality and design complexity.

**Interpretation Variability:** EarthScape relies on expert-labeled SG maps. Classes are well-defined by geologic process, but boundaries may be approximate where diagnostic features are sparse, introducing uncertainty into patch-level labels. In our benchmarks, a unit is marked as present if it occupies at least one pixel. We provide per-patch class-area proportions to support alternative thresholding or probabilistic labeling.

**Label and Taxonomy Constraints:** The current release uses a single, aggregated taxonomy of seven SG units and does not capture the full diversity of SG materials observed globally. This limits the breadth of environments represented and may constrain the generality of models trained solely on EarthScape v1.0. At the same time, classes are defined in terms of surface process, enabling broad transferability to regions with similar geologic processes and data.

**Temporal Inconsistency:** Input modalities were acquired from 2019 to 2024, introducing mild temporal misalignment among imagery, elevation, and vector layers. While SG units are stable on these timescales, land cover and infrastructure may change, creating minor label noise. This asynchrony is a limitation, but also reflects realistic conditions under which many Earth observation systems operate.

**Patch Overlap and Sample Independence:** EarthScape uses a 50% overlapping patches to increase spatial context, ensure dense sampling, and support multi-view aggregation, but this design also introduces statistical dependence between neighboring samples. We mitigate leakage in evaluation by enforcing spatially disjoint train/validation/test sets, but non-independence remains a consideration when designing models and interpreting significance.

# 6 CONCLUSIONS

We introduced EarthScape, an AI-ready multimodal benchmark for SG mapping. EarthScape integrates aerial imagery, DEMs, multi-scale terrain derivatives, and GIS vector layers into a unified, co-registered framework, providing a modality-deep testbed for surface-aware geospatial learning. The dataset exposes real-world challenges that are underrepresented in existing benchmarks, including long-tailed class distributions, multi-label patch structure, multi-scale organization, and explicit geographic domain shift between training and held-out regions.

In our baseline experiments, terrain-derived features that encode surface shape emerge as the most informative and robust modalities, while models relying primarily on RGB or raw elevation suffer substantial degradation under cross-region evaluation. Multi-scale and multimodal inputs improve performance over single-scale or single-modality configurations. Cross-region transfer is more sensitive to how surface inputs are fused than to backbone encoder complexity, with early channel stacking consistently outperforming attention-based fusion. SGMap-Net is a lightweight baseline, yet outperforms the recent spectral-based RS foundation models we evaluate. These findings underscore that SG mapping in EarthScape is strongly shape-driven and indicate limits on the transferability of appearance-based representations in this setting.

EarthScape is designed as a living, versioned dataset and will expand in both geographic coverage and modality space as high-quality SG maps and compatible remote sensing products become available and pass our quality-control pipeline. By releasing all data, code, and benchmark splits, we aim to support reproducible research on multimodal fusion, domain adaptation, and geospatial learning, and to provide a common platform for cross-disciplinary work at the intersection of computer vision, RS, and Earth surface analysis.

ETHICS STATEMENT

This work adheres to the ICLR Code of Ethics. EarthScape is built exclusively from publicly available, government or community datasets under open licenses; no human subjects, personal data, or sensitive information are involved. All source attributions and licensing terms are respected, and no conflicts of interest are present. We caution that models trained on EarthScape should be applied with geological domain expertise, particularly outside regions with similar surficial processes, to avoid misinterpretation in decision-making contexts. We report implementation details in the Appendix to promote awareness of environmental impact and enable informed replication.

REPRODUCIBILITY STATEMENT

We support reproducibility through precise documentation of data sources and preprocessing, patch generation and spatially independent splits, model and training configurations, and comprehensive results. Upon acceptance, the full EarthScape dataset and code will be publicly released with a data dictionary and README. These materials are intended to allow end-to-end reproduction of all reported experiments.

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

## A EarthScape Details

### A.1 Purpose

EarthScape is designed as a benchmark dataset for learning from continuous, spatially coherent SG units and the surface processes they represent. Its primary purpose is to support research on multi-modal geospatial learning, where models integrate aerial imagery, LiDAR-derived DEMs, multi-scale terrain derivatives, and vector contextual data to infer geologic patterns expressed on the Earth's surface. The name EarthScape reflects this focus on surface morphology and near-surface processes, rather than implying complete global coverage.

### A.2 Code Availability and Reproducibility

All code used for data preprocessing, patch generation, model training, and evaluation will be released upon acceptance. The repository will include comprehensive documentation and scripts to fully reproduce the dataset and all experiments reported in this contribution. This includes tools for multimodal data alignment, terrain-derivative computation, mask rasterization, and construction of spatially independent training/validation/test splits. The codebase also provides baseline implementations of SGMap-Net with both ResNeXt-50 and ViT-B/16 encoder backbones, along with standardized training and evaluation pipelines. Utilities for focal loss, threshold optimization, performance metrics, and visualization are included for completeness. The full dataset will also be made publicly available at acceptance. The dataset archive contains all co-registered modalities, multilabel target masks, per-patch class proportions, and accompanying metadata, including a detailed data dictionary documenting each modality.

### A.3 Dataset Contents

EarthScape provides a standardized multimodal dataset for each $256{\times}256$ patch aligned to a common 1.52 m GSD grid in the EPSG:3089 coordinate reference system. For every patch, the dataset includes co-registered raster modalities (RGB, DEM, multi-scale EP, PrC, PlC, S, and SDS terrain derivatives), along with binary hydrology (NHD) and infrastructure (OSM) masks. Each patch is paired with a multilabel one-hot vector for the seven surficial geologic units, per-class area proportions, and a GeoJSON polygon defining the exact patch footprint and unique patch ID. All rasters are provided as GeoTIFF files, labels and areas as CSV, and patch polygons as vector GeoJSON files. The dataset archive additionally includes global normalization statistics (per-modality means and standard deviations) computed over the full in-domain region to support reproducible preprocessing. Table 3 summarizes all contents included in the current dataset.

### A.4 Current Status and Roadmap

Figure 4 illustrates the current extent and planned expansion of the EarthScape dataset. EarthScape v1.0 includes two regions in central Kentucky: Warren County, which contains the largest number of image patches, and Hardin County, which serves as an independent test area that enables evaluation of cross-region generalization. Version 2.0 will nearly triple the number of patches (Fig. 4), while Version 3.0 will extend coverage beyond Kentucky into adjacent regions that capture additional geologic processes and environmental conditions. EarthScape is designed as a living dataset. Future versions will continue to evolve through the addition of new regions, modalities, and metadata. We invite external researchers to contribute high-quality data that aligns with the dataset's standards, with the goal of strengthening EarthScape as a shared benchmark for multimodal geospatial learning.

### A.5 Extensibility and Community Contributions

EarthScape is designed as a living dataset rather than a one-time release. To maintain reproducibility while enabling growth, we follow semantic versioning with frozen releases (v1.0, v1.1, v2.0, etc.), stable train/validation/test splits, and a public CHANGELOG documenting all modifications to regions, modalities, or preprocessing steps. Newly added areas are organized as separate modules so that existing benchmarks remain stable across versions.

Table 3: Summary of EarthScape v1.0 dataset contents.

| Name | Filename Pattern | Data Type | Metadata |
|---|---|---|---|
| Mask | {id}_geology.tif | float | SG target mask for segmentation; 1.52 m GSD |
| DEM | {id}_dem.tif | float | Airborne LiDAR; 1.52 m GSD |
| Aerial, Red | {id}_aerialr.tif | float | Aerial imagery, red band; 1.52 m GSD |
| Aerial, Green | {id}_aerialg.tif | float | Aerial imagery, green band; 1.52 m GSD |
| Aerial, Blue | {id}_aerialb.tif | float | Aerial imagery, blue band; 1.52 m GSD |
| Aerial, NIR | {id}_aerialr.tif | float | Aerial imagery, near infrared band; 1.52 m GSD |
| Hydrography | {id}_nhd.tif | float | Binary stream & water bodies; 1.52 m GSD |
| Infrastructure | {id}_osm.tif | float | Binary road & railways; 1.52 m GSD |
| $EP_5$ | {id}_ep_5x5.tif | float | Computed with $5\times5$ kernel & 1.52 m GSD DEM |
| $EP_{11}$ | {id}_ep_11x11.tif | float | Computed with $11\times11$ kernel & 1.52 m GSD DEM |
| $EP_{21}$ | {id}_ep_21x21.tif | float | Computed with $21\times21$ kernel & 1.52 m GSD DEM |
| $EP_{51}$ | {id}_ep_51x51.tif | float | Computed with $51\times51$ kernel & 1.52 m GSD DEM |
| $EP_{101}$ | {id}_ep_101x101.tif | float | Computed with $101\times101$ kernel & 1.52 m GSD DEM |
| $EP_{201}$ | {id}_ep_201x201.tif | float | Computed with $201\times201$ kernel & 1.52 m GSD DEM |
| $PlC_{1.5}$ | {id}_plancurv.tif | float | Computed with $5\times5$ kernel & 1.52 m GSD DEM |
| $PlC_3$ | {id}_plancurv_10.tif | float | Computed with $5\times5$ kernel & 3.05 m GSD DEM |
| $PlC_6$ | {id}_plancurv_20.tif | float | Computed with $5\times5$ kernel & 6.1 m GSD DEM |
| $PlC_{15}$ | {id}_plancurv_50.tif | float | Computed with $5\times5$ kernel & 15.24 m GSD DEM |
| $PlC_{30}$ | {id}_plancurv_100.tif | float | Computed with $5\times5$ kernel & 30.48 m GSD DEM |
| $PlC_{60}$ | {id}_plancurv_200.tif | float | Computed with $5\times5$ kernel & 60.96 m GSD DEM |
| $PrC_{1.5}$ | {id}_procurv.tif | float | Computed with $5\times5$ kernel & 1.52 m GSD DEM |
| $PrC_3$ | {id}_procurv_10.tif | float | Computed with $5\times5$ kernel & 3.05 m GSD DEM |
| $PrC_6$ | {id}_procurv_20.tif | float | Computed with $5\times5$ kernel & 6.1 m GSD DEM |
| $PrC_{15}$ | {id}_procurv_50.tif | float | Computed with $5\times5$ kernel & 15.24 m GSD DEM |
| $PrC_{30}$ | {id}_procurv_100.tif | float | Computed with $5\times5$ kernel & 30.48 m GSD DEM |
| $PrC_{60}$ | {id}_procurv_200.tif | float | Computed with $5\times5$ kernel & 60.96 m GSD DEM |
| $S_{1.5}$ | {id}_slope.tif | float | Computed with $5\times5$ kernel & 1.52 m GSD DEM |
| $S_3$ | {id}_slope_10.tif | float | Computed with $5\times5$ kernel & 3.05 m GSD DEM |
| $S_6$ | {id}_slope_20.tif | float | Computed with $5\times5$ kernel & 6.1 m GSD DEM |
| $S_{15}$ | {id}_slope_50.tif | float | Computed with $5\times5$ kernel & 15.24 m GSD DEM |
| $S_{30}$ | {id}_slope_100.tif | float | Computed with $5\times5$ kernel & 30.48 m GSD DEM |
| $S_{60}$ | {id}_slope_200.tif | float | Computed with $5\times5$ kernel & 60.96 m GSD DEM |
| $SDS_5$ | {id}_stdslope_5x5.tif | float | Computed with $5\times5$ kernel & 1.52 m GSD DEM |
| $SDS_{11}$ | {id}_stdslope_11x11.tif | float | Computed with $11\times11$ kernel & 1.52 m GSD DEM |
| $SDS_{21}$ | {id}_stdslope_21x21.tif | float | Computed with $21\times21$ kernel & 1.52 m GSD DEM |
| $SDS_{51}$ | {id}_stdslope_51x51.tif | float | Computed with $51\times51$ kernel & 1.52 m GSD DEM |
| $SDS_{101}$ | {id}_stdslope_101x101.tif | float | Computed with $101\times101$ kernel & 1.52 m GSD DEM |
| $SDS_{201}$ | {id}_stdslope_201x201.tif | float | Computed with $201\times201$ kernel & 1.52 m GSD DEM |
| Class Areas | earthscape_areas.csv | float | Patch-level class-area proportions |
| Labels | earthscape_labels.csv | int | One-hot encoded labels (no pixel threshold) |
| Patch GIS | earthscape_patches.geojson | - | Vector file with locations & geometries |
| Statistics | earthscape_stats.csv | float | Modality mean & SDs from training split |
| Mapping | earthscape_class_mapping.json | - | Label string to ordinal mapping |
| Train Split | indomain_train.geojson | - | Training split GIS file with patch IDs |
| Val. Split | indomain_val.geojson | - | Validation split GIS file with patch IDs |
| In-dom. Test Split | indomain_test.geojson | - | In-domain test split GIS file with patch IDs |
| Cross-dom. Test Split | crossdomain_test.geojson | - | Cross-domain test split GIS file with patch IDs |

Although the preprocessing pipeline is fully implemented, incorporating additional SG maps requires coordinated domain and data-engineering effort. Each new region must be standardized with EarthScape's process-based SG classes, rasterized with topologically consistent masks, aligned with LiDAR-quality DEMs and imagery, and evaluated for geologic validity remaining uncertainty. External groups may propose new regions by providing high-quality 1:24,000-scale SG maps together with co-registered DEMs, terrain derivatives, aerial imagery, and relevant vector layers. Regions meeting EarthScape's quality standards and QC protocol will be incorporated into a subsequent versioned release.

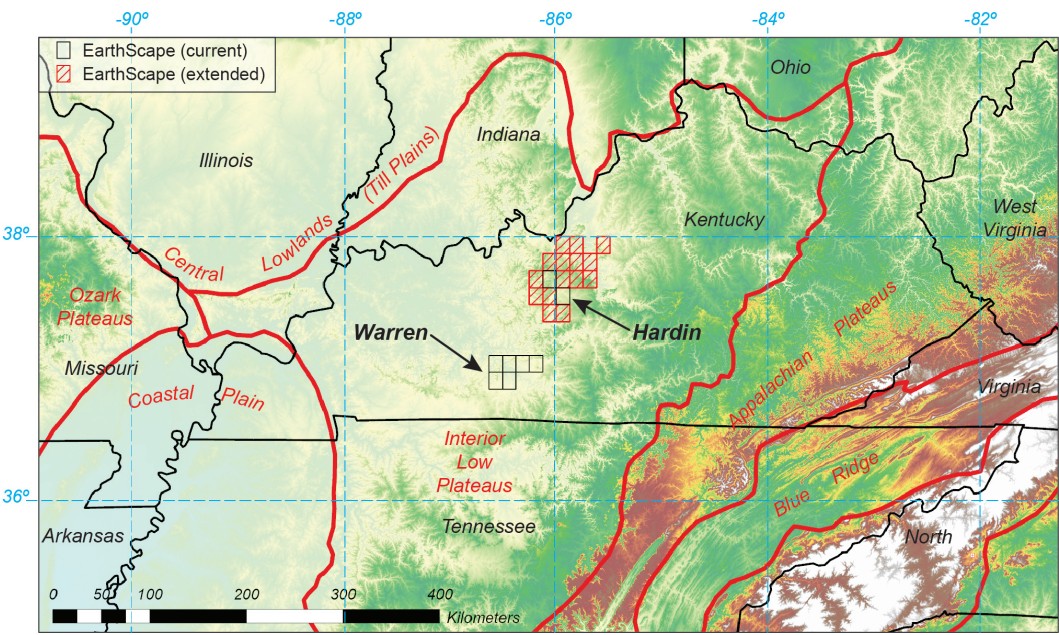

Figure 4: Map of the central United States showing the publicly available 1:24,000-scale surficial geologic maps. Red lines show boundaries of major geologic provinces, which provide geological constraints for generalizability. EarthScape-trained models are expected to generalize effectively throughout the Interior Low Plateaus and adjacent Appalachian Plateaus, based on shared terrain, bedrock, and geomorphic processes. In contrast, the glaciated Central Lowlands and Coastal Plain are characterized by fundamentally different surficial processes and materials.

## B  GEOLOGICAL BACKGROUND AND TERRAIN DERIVATIVES

### B.1  SURFICIAL GEOLOGY

Figure 5 presents two examples of SG maps from the EarthScape dataset, shown as semi-transparent overlays atop multi-directional hillshade images. This visualization emphasizes the relationship between SG and topography. Distinct landforms, such as river valleys, plains, and steep hillslopes, are spatially correlated with specific surficial geologic units. EarthScape leverages this relationship to frame surficial geologic mapping as a vision task, where computer vision models can learn to associate surface patterns with underlying geological processes. The EarthScape dataset currently includes seven surficial geologic map units, each representing distinct surface processes (Table 4). Although the maps are from Kentucky, the units reflect fluvial deposition, gravitational transport, and in-situ weathering processes that are active in many landscapes worldwide.

1. *Artificial fill (af1):* Manmade deposits consisting of transported or excavated material placed or removed for engineering, mining, or other anthropogenic structures. Includes road embankments, building pads, quarries, and areas of significant topographic modification. Often exhibits sharp, angular boundaries. The spatial extent of af1 can be below the mapping resolution and inconsistently captured on expert-curated surficial geologic maps.

2. *Alluvium (Qal):* Unconsolidated sediments, typically consisting of clay-, silt-, sand-, and gravel-sized particles, deposited by modern rivers and streams. Qal is commonly found in active floodplains and valley bottoms and reflects recent sedimentation from overbank flooding and channel migration. These areas are generally flat, vegetated, and hydrologically dynamic.

3. *Alluvial fans (Qaf):* Fan-shaped deposits formed at the base of tributaries or drainages, where sediment-laden water rapidly spreads and loses energy. These deposits are typically coarse-grained, poorly sorted, and associated with debris flows or flash floods. Although geologically significant, Qaf are often small, making them inconsistently represented on typical 1:24,000-scale maps.

4. *Terrace deposits (Qat):* Relict alluvial sediments preserved on elevated flat surfaces above modern stream channels. These deposits reflect former floodplain levels and subsequent stream incision. Compositionally similar to Qal, but usually expressed as distinct landforms above modern flood plains.

5. *Colluvium (Qc):* Hillslope-derived sediments that accumulate at the base of slopes due to gravity-driven processes such as soil creep, slopewash, and shallow landslides. Qc deposits are unsorted and variable in thickness, typically found on slopes $> 12°$. Qc is considered an active geomorphic unit.

6. *Colluvial aprons (Qca):* Slope-derived material deposited across lower hillslopes. Qca typically occurs downslope from Qc and is more stable, having accumulated over longer time periods. These deposits may be partially weathered, with poorly defined lower boundaries that grade into Qr due to extended weathering and lower erosion rates.

7. *Residuum (Qr):* Weathered material formed in place from the physical, chemical, and biological breakdown of underlying bedrock or older unconsolidated deposits. Qr lacks significant sediment transportation and is commonly found in upland areas with minimal active erosion. Qr is commonly gradational and poorly defined where it grades into Qc or Qca, leading to interpretive ambiguity during mapping.

### B.2  GEOLOGIC GENERALIZATION

Although EarthScape v1.0 is geographically limited, the geologic processes and terrain surface types it represents are not unique. The dataset is directly applicable to the surficial geology exposed in the Interior Low Plateaus and Appalachian Plateaus (Fig. 4). Comparable landscapes characterized by carbonate bedrock, dissected plains, and mixed fluvial–colluvial systems occur globally, including the Ozark Plateau (USA), parts of the Carpathians (Eastern Europe), the Dinaric Alps (Balkans), and areas of central China and southeastern Australia. However, differences in geologic processes do constrain transferability. For instance, the Central Lowlands (Fig. 4) contain fundamentally different

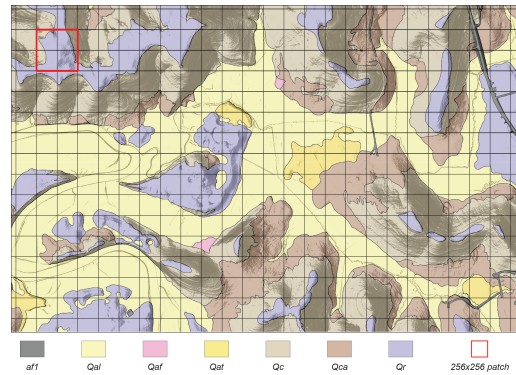 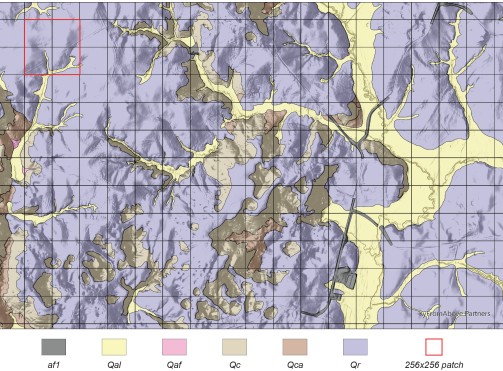

(a) Surficial geologic map of part of Warren County.  (b) Surficial geologic map of part of Hardin County.

Figure 5: Example SG maps showing the distribution of unconsolidated materials overlaid on hillshade images to emphasize topographic context. The spatial correspondence between SG map units and landscape features, such as valleys and slopes, is visually apparent. The black grid indicates the layout of EarthScape patches, each measuring $256 \times 256$ pixels ($390.14 \times 390.14$ m) with 50% overlap. Red squares in the upper left corners highlight a single patch

Table 4: Descriptions of SG units represented in EarthScape v1.0.

| Class | Name | Dominant Process | Visual Cues |
|-------|------|------------------|-------------|
| af1 | Artificial fill | Anthropogenic | Sharp, angular edges; linear or rectilinear shapes; DEM anomalies inconsistent with natural terrain. |
| Qal | Alluvium | Water-dominated | Relatively wide, flat-bottomed valleys; active stream channels; low relative elevations. |
| Qaf | Alluvial fans | Water-dominated (acute) | Small, isolated, lobate landforms; located at slope-base transitions. |
| Qat | Terrace deposits | Water-dominated (relict) | Flat benches above floodplains; stepped margins; often dissected. |
| Qc | Colluvium | Gravity-dominated (active) | Steep slopes ($> 12°$); may include landslides or erosional hazards. |
| Qca | Colluvial aprons | Gravity-dominated (stable) | Wedge-shaped landforms along slope bases with concave profiles; transitional between slope and plain. |
| Qr | Residuum | In-situ weathering | Broad, low-relief uplands; little drainage or erosion; variable surface texture. |

surficial materials and geomorphic processes as a result of widespread glaciation (rather than non-glaciated weathering and erosion), limiting the direct applicability of EarthScape v1.0. Accordingly, we recommend that applications of EarthScape v1.0 to new regions be guided by domain expertise to ensure geological validity and meaningful interpretation.

### B.3 MODALITIES

Figs. 6 and 7 showcase the diverse, multimodal data available for each of the 31,018 EarthScape patches. Each patch includes 38 co-registered channels, comprising expert-labeled geologic masks, high-resolution aerial RGB and NIR imagery, a DEM, terrain features derived from the DEM at multiple spatial scales, and rasterized vector data representing hydrologic and infrastructure features. Among these modalities, the DEM and its derived terrain features provide critical context for understanding surface processes and interpreting surficial geologic units. Five terrain variables were computed at six spatial scales to capture localized and regional landform variability.

1. *Slope (S)* is the first derivative of elevation, measuring the rate of change of elevation over a horizontal distance. It quantifies the steepness of the terrain, providing insight into processes like erosion and material movement.

$$S = \tan^{-1} \left( \sqrt{\left(\frac{\partial z}{\partial x}\right)^2 + \left(\frac{\partial z}{\partial y}\right)^2} \right) \tag{3}$$

Where $\frac{\partial z}{\partial x}$ and $\frac{\partial z}{\partial y}$ are the partial derivatives of elevation in the x and y directions, respectively.

2. *Profile curvature (PrC)* is a directional second derivative of elevation, measured along the direction of the steepest slope. It quantifies how slope changes in that direction, reflecting the acceleration or deceleration of flow, and influencing erosion and deposition patterns.

$$PrC = \frac{p^2 r + 2pqs + q^2 t}{(p^2 + q^2)^{3/2}} \tag{4}$$

Where $p = \frac{\partial z}{\partial x}$ and $q = \frac{\partial z}{\partial y}$ are the first-order partial derivatives of elevation in the x and y directions, and $r = \frac{\partial^2 z}{\partial x^2}$, $s = \frac{\partial^2 z}{\partial x \partial y}$, and $t = \frac{\partial^2 z}{\partial y^2}$ are the corresponding second-order partial derivatives.

3. *Planform curvature (PlC)* is another directional second derivative of elevation, measured perpendicular to the direction of the steepest slope. It describes the curvature of contour lines (lines of equal elevation) and reflects how flow paths converge or diverge across the landscape.

$$PlC = \frac{q^2 r - 2pqs + p^2 t}{(p^2 + q^2)^{3/2}} \tag{5}$$

Where $p = \frac{\partial z}{\partial x}$ and $q = \frac{\partial z}{\partial y}$ are the first-order partial derivatives of elevation in the x and y directions, and $r = \frac{\partial^2 z}{\partial x^2}$, $s = \frac{\partial^2 z}{\partial x \partial y}$, and $t = \frac{\partial^2 z}{\partial y^2}$ are the corresponding second-order partial derivatives.

4. *Elevation percentile (EP)* measures the relative elevation of a point within a defined neighborhood, expressed as a percentile rank (0–100%) of the elevation among neighboring values. EP helps distinguish between landforms defined by relative topography, such as ridges, valleys, or sinkholes.

$$EP = 100 \cdot \frac{|\{z_i \in Z \mid z_i < z\}|}{N} \tag{6}$$

Where $z$ is the elevation at the center cell, $Z$ is the set of elevations in the neighborhood, $z_i$ are the individual neighboring elevations, and $N$ is the total number of neighbors. The numerator counts the number of neighbors with elevation less than $z$.

5. *Standard deviation of slope (SDS)* is a measure of roughness and quantifies the variability in slope angle within a local window. SDS represents how rugged or uneven the surface is, highlighting areas with complex topography that may correlate with diverse geologic materials or processes.

$$SDS = \sqrt{\frac{1}{N} \sum_{i=1}^{N} \left(S_i - \bar{S}\right)^2} \tag{7}$$

Where $S_i$ is the slope angle (in degrees or radians) of the $i^{th}$ cell in the neighborhood, $\bar{S}$ is the mean slope within that neighborhood, and $N$ is the total number of cells used in the calculation window.

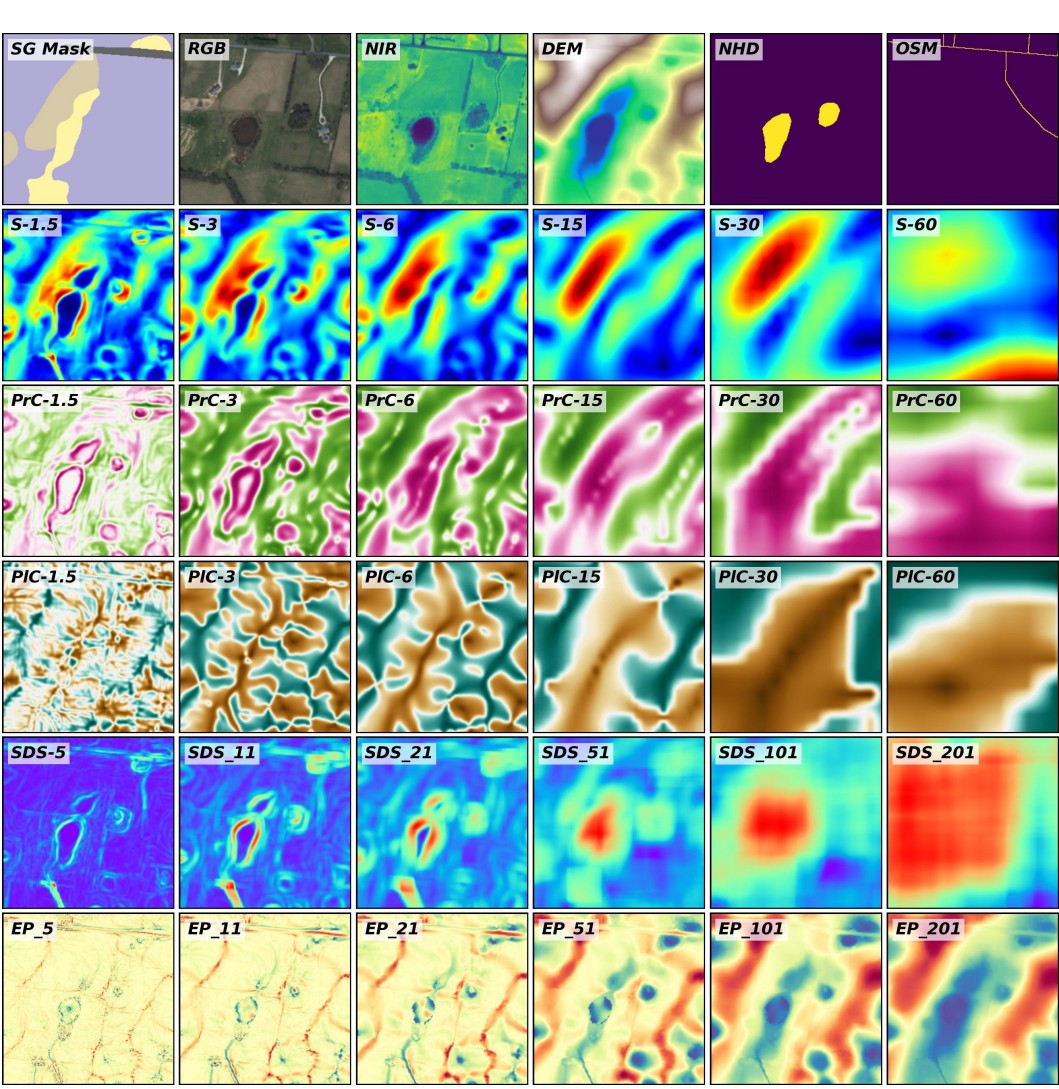

Figure 6: Example patch from the Warren County area showcasing the 38 channels available in EarthScape. Channels are displayed from top left to bottom right: target mask, RGB aerial imagery, NIR aerial imagery, DEM, NHD hydrologic features, OSM infrastructure, six spatial scales of S, PrC, and PlC derived from downsampled DEMs, and multiple scales of SDS and EP calculated using six kernel sizes with the original DEM.

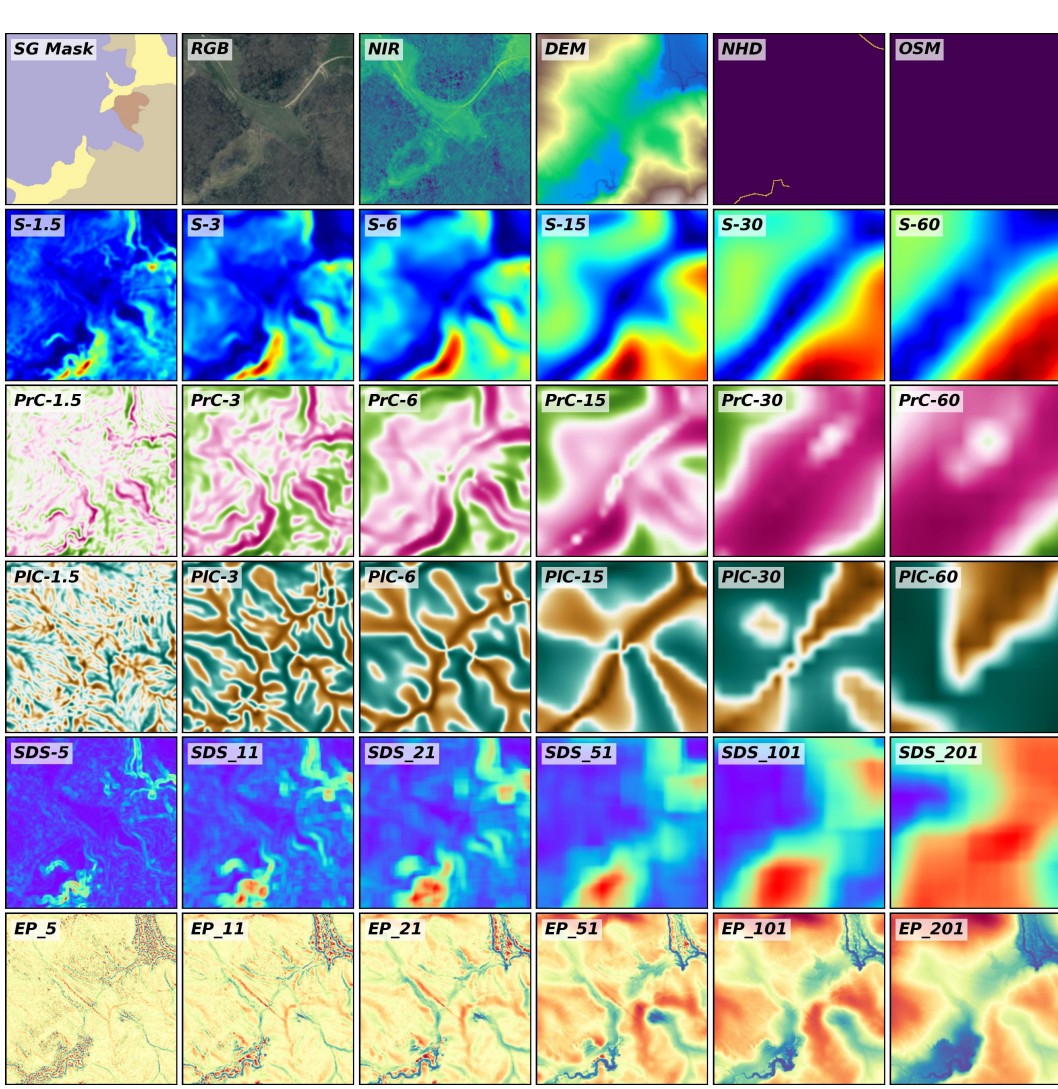

Figure 7: Example patch from the Hardin County area showcasing the 38 channels available in EarthScape. Channels are displayed from top left to bottom right: target mask, RGB aerial imagery, NIR aerial imagery, DEM, NHD hydrologic features, OSM infrastructure, six spatial scales of S, PrC, and PlC derived from downsampled DEMs, and multiple scales of SDS and EP calculated using six kernel sizes with the original DEM.

## C  ADDITIONAL BENCHMARK DETAILS

### C.1  GEOSPATIAL PATCH SELECTION AND EXPERIMENTAL DESIGN

To ensure robust and geographically fair model evaluation, EarthScape patches were split into spatially independent training, validation, and test sets. The Warren County region was used for in-domain training and evaluation due to its broader spatial coverage and diversity of surficial geologic units. We first randomly selected 1,536 test patches, followed by 768 validation patches that did not spatially intersect with the test set, and then assigned the remaining 8,416 non-overlapping patches to the training set (Fig. 8). These split sizes were chosen through iterative selection to satisfy several practical constraints: (1) all splits had to be spatially non-overlapping; (2) patch counts needed to be divisible by common batch sizes (e.g., 16 or 32) to support efficient model training; (3) the resulting proportions had to be reasonably balanced and typical for supervised learning workflows (Table 5).

To assess geographic generalization, we created a cross-domain test set consisting of 1,536 randomly selected patches from the Hardin County region (Fig. 8). Although geologically similar, Hardin County is located approximately 85 km from Warren County and is spatially independent. This separate region enables testing model performance under domain shift, simulating real-world conditions in which models are applied beyond the area used for training.

Figure 9 shows the class distributions for each data split. All subsets reflect the inherent class imbalance typical of surficial geologic mapping, driven by the localized nature of surface processes. Importantly, the class distributions are consistent across the training, validation, and both test sets, ensuring that evaluation performance is not biased by differences in class representation.

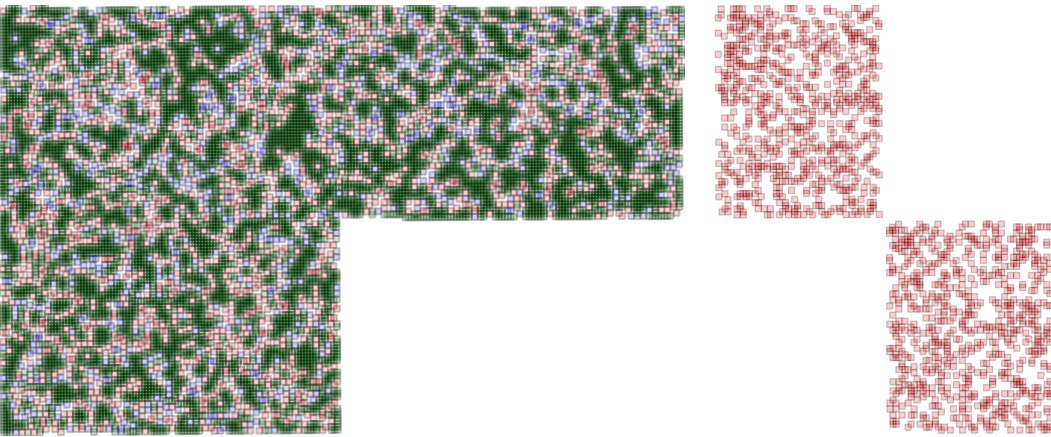

(a) Training, validation, and in-domain test patches from the Warren County region.

(b) Cross-domain test patches from the Hardin County region.

Figure 8: Spatial distribution of selected patches for EarthScape experiments. All splits are spatially independent: no patch overlaps between splits, though patches within the same split may partially overlap due to the 50% patch stride. See Figure 4 for geographic locations.

Table 5: Patch counts and split proportions for training, validation, and testing based on the total number of patches used for in-domain training and evaluation. An additional test set from the spatially independent Hardin County region was used to assess cross-domain generalization.

| Split | Region | Patch Count ($n$) | In-domain Proportion (%) |
|---|---|---|---|
| Training | Warren | 8,416 | 78.5 |
| Validation | Warren | 768 | 7.2 |
| In-domain Testing | Warren | 1,536 | 14.3 |
| Cross-domain Testing | Hardin | 1,536 | - |

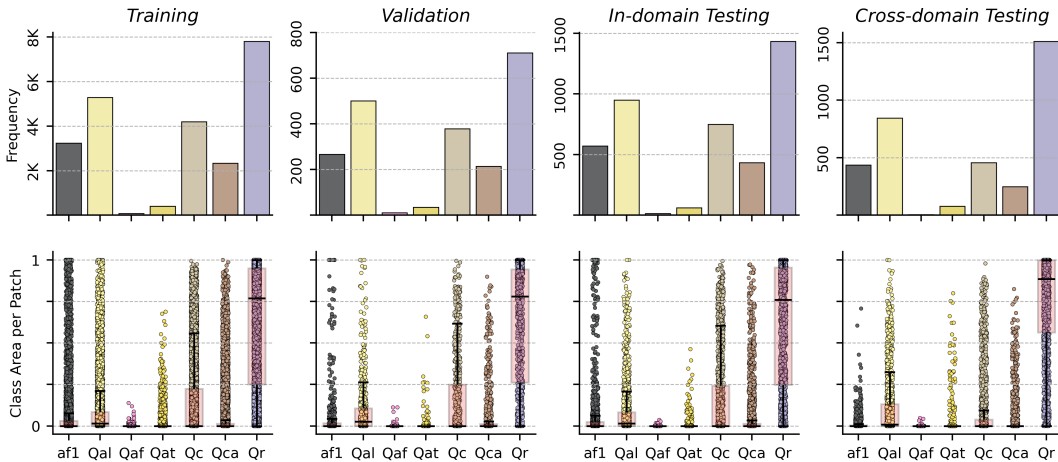

Figure 9: Class distribution and intra-patch composition across EarthScape data splits. Top row: Bar plots showing the frequency of each surficial geologic unit in the training, validation, in-domain test, and cross-domain test sets. Bottom row: Swarm plots overlaid with box plots showing the proportion of each patch occupied by each class. All splits display consistent patterns in both overall frequency and within-patch composition, supporting fair evaluation across subsets.

## C.2 HARDWARE, COMPUTE, AND TRAINING CONFIGURATION

All experiments were implemented in Python using the PyTorch framework. Models were trained and evaluated on a machine equipped with an Intel Xeon processor, 128 GB of RAM, and two NVIDIA RTX A4000 GPUs. Initial training experiments were run for 25 epochs to observe convergence behavior (Fig. 10). For any single-channel configuration (e.g., DEM-only), SGMap-Net with the ResNeXt-50 encoder contains 25.35 M trainable parameters and requires 5.56 GFLOPs per $256 \times 256$ forward pass, while the ViT-B/16 encoder variant contains 87.51 M trainable parameters and requires 16.87 GFLOPs. FLOPs increase slightly when multiple modalities are included, but parameter count is invariant. Across all configurations, we found that model performance generally stabilized within the first 10 epochs of training (Fig. 10). Based on these observations, we standardized all subsequent experiments to 15 epochs, which provided a balance between sufficient training and computational efficiency.

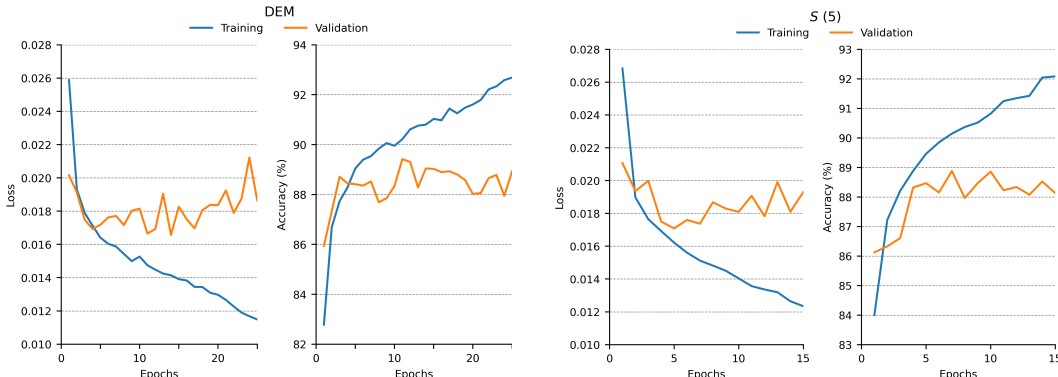

(a) DEM model trained for 25 epochs. Early convergence is evident by epoch 10, with decreased performance thereafter.

(b) $S$ (5) model trained for 15 epochs, demonstrating stable convergence and alignment between training and validation performance.

Figure 10: Training and validation loss and accuracy curves across epochs. Each subplot shows model loss (left panel) and accuracy (right panel) behavior for a different input modality, with training curves shown in blue and validation curves in orange.

### C.3 FOCAL LOSS

To address the significant class imbalance in EarthScape, we adopted focal loss. Initial tuning was conducted using the validation set and DEM modality only, a ResNeXt-50 backbone, the Adam optimizer, and a fixed learning rate of 0.001 to explore the effects of focal loss parameters. We evaluated values of $\gamma \in 1.0,\ 1.5,\ 2.0,\ 2.5,\ 3.0$ and tested several strategies for the class-balancing factor ($\alpha$), including a fixed scalar ($\alpha = 0.25$), inverse class frequency (ICF), square root of ICF ($\sqrt{\text{ICF}}$), and class-balanced focal loss with $\beta = 0.999$ (CBFL) (Table 6). The combination of $\alpha = \sqrt{\text{ICF}}$ and $\gamma = 2.0$ yielded the best performance for the DEM-only configuration. However, when this setting was applied to other modalities, training became unstable, and convergence was inconsistent. To ensure comparability across all experiments and isolate the effects of modality and fusion design, we adopted the original focal loss settings ($\alpha = 0.25, \gamma = 2.0$) for all remaining runs.

Table 6: Per-class and macro-averaged validation set F1 and AUC scores for different focal loss configurations using the DEM modality and a ResNeXt-50 backbone. These results were used to guide focal loss tuning, although the best-performing configuration did not generalize well across modalities. As a result, we adopted $\alpha = 0.25, \gamma = 2.0$ for all subsequent experiments.

| $\alpha$ | $\gamma$ | F1 | | | | | | | | AUC | | | | | | | |
|---|---|---|---|---|---|---|---|---|---|---|---|---|---|---|---|---|---|
| | | af1 | Qal | Qaf | Qat | Qc | Qca | Qr | AVG. | af1 | Qal | Qaf | Qat | Qc | Qca | Qr | AVG. |
| 0.25 | 1 | 0.743 | 0.848 | 0.267 | 0.436 | 0.899 | 0.778 | 0.968 | 0.706 | 0.861 | 0.862 | 0.907 | 0.923 | 0.967 | 0.923 | 0.937 | 0.911 |
| 0.25 | 1.5 | 0.726 | 0.855 | 0.250 | 0.354 | 0.914 | 0.751 | 0.968 | 0.688 | 0.866 | 0.874 | 0.915 | 0.884 | 0.964 | 0.909 | 0.932 | 0.906 |
| 0.25 | 2 | 0.749 | 0.841 | 0.229 | 0.400 | 0.914 | 0.778 | 0.965 | 0.697 | 0.868 | 0.859 | 0.929 | 0.919 | 0.970 | 0.929 | 0.912 | 0.912 |
| 0.25 | 2.5 | 0.690 | 0.866 | 0.275 | 0.387 | 0.895 | 0.767 | 0.971 | 0.693 | 0.844 | 0.887 | 0.944 | 0.895 | 0.965 | 0.920 | 0.945 | 0.914 |
| 0.25 | 3 | 0.709 | 0.851 | 0.267 | 0.323 | 0.890 | 0.772 | 0.970 | 0.683 | 0.853 | 0.863 | 0.895 | 0.890 | 0.962 | 0.925 | 0.924 | 0.902 |
| ICF | 1 | 0.524 | 0.804 | 0.204 | 0.390 | 0.831 | 0.640 | 0.961 | 0.622 | 0.639 | 0.730 | 0.921 | 0.851 | 0.912 | 0.828 | 0.851 | 0.819 |
| ICF | 2 | 0.596 | 0.805 | 0.286 | 0.314 | 0.839 | 0.687 | 0.961 | 0.641 | 0.731 | 0.737 | 0.934 | 0.828 | 0.916 | 0.854 | 0.869 | 0.838 |
| ICF | 2.5 | 0.589 | 0.799 | 0.267 | 0.326 | 0.843 | 0.671 | 0.962 | 0.637 | 0.711 | 0.716 | 0.923 | 0.838 | 0.919 | 0.842 | 0.848 | 0.828 |
| $\sqrt{\text{ICF}}$ | 1 | 0.696 | 0.845 | 0.286 | 0.348 | 0.879 | 0.763 | 0.965 | 0.683 | 0.843 | 0.867 | 0.912 | 0.905 | 0.955 | 0.925 | 0.922 | 0.904 |
| $\sqrt{\text{ICF}}$ | 1.5 | 0.688 | 0.838 | 0.333 | 0.409 | 0.877 | 0.766 | 0.974 | 0.698 | 0.834 | 0.844 | 0.961 | 0.909 | 0.951 | 0.914 | 0.924 | 0.905 |
| $\sqrt{\text{ICF}}$ | 2 | 0.726 | 0.841 | 0.444 | 0.460 | 0.905 | 0.749 | 0.962 | 0.727 | 0.850 | 0.853 | 0.945 | 0.931 | 0.961 | 0.921 | 0.913 | 0.911 |
| $\sqrt{\text{ICF}}$ | 2.5 | 0.709 | 0.835 | 0.293 | 0.487 | 0.901 | 0.760 | 0.963 | 0.707 | 0.849 | 0.844 | 0.956 | 0.940 | 0.962 | 0.926 | 0.893 | 0.910 |
| CBFL | 1 | 0.720 | 0.831 | 0.412 | 0.427 | 0.893 | 0.733 | 0.973 | 0.713 | 0.864 | 0.839 | 0.965 | 0.903 | 0.962 | 0.902 | 0.924 | 0.908 |
| CBFL | 1.5 | 0.715 | 0.841 | 0.286 | 0.412 | 0.908 | 0.764 | 0.971 | 0.700 | 0.844 | 0.854 | 0.940 | 0.906 | 0.971 | 0.920 | 0.947 | 0.912 |
| CBFL | 2 | 0.727 | 0.866 | 0.357 | 0.455 | 0.914 | 0.792 | 0.965 | 0.725 | 0.867 | 0.890 | 0.918 | 0.923 | 0.971 | 0.921 | 0.914 | 0.915 |
| CBFL | 2.5 | 0.711 | 0.844 | 0.455 | 0.372 | 0.911 | 0.753 | 0.968 | 0.716 | 0.846 | 0.857 | 0.970 | 0.908 | 0.967 | 0.928 | 0.930 | 0.915 |

### C.4 MAXIMUM MEAN DISCREPANCY ANALYSIS

To quantify cross-region distributional differences between Warren and Hardin, we compute the maximum mean discrepancy (MMD) between patch-level feature distributions (Gretton et al., 2012). Each 256×256 patch is summarized using the 10th, 25th, 50th, 75th, and 90th percentiles of pixel intensities for the relevant modality. For multi-channel inputs, percentile features are concatenated into a joint feature vector. Percentile vectors from both regions are pooled and scaled to $[0, 1]$, then compared using RBF-kernel MMD. Table 7 reports MMD values for representative modalities. These values indicate measurable, modality-specific covariate shift between regions, reflecting differences in appearance, elevation, and multi-scale terrain structure.

Table 7: MMD for selected raw inputs in EarthScape v1.0.

| Modality | MMD |
|---|---|
| RGB | 0.3654 |
| DEM | 0.8322 |
| $EP_{51}$ | 0.2438 |
| $S_{1.5}$ | 0.0974 |
| $SDS_{21}$ | 0.0775 |
| $S_{ms}$ | 0.1549 |
| $EP_{ms}+S_{ms}+SDS_{ms}$ | 0.1636 |

# D COMPREHENSIVE RESULTS

## D.1 SINGLE MODALITY

Tables 8, 9, and 10 report complete results for all single-scale, single-modality experiments, including macro-averaged F1, AUC, precision, recall, mean average precision (mAP), and accuracy for both the in-domain and cross-domain evaluations. Results are provided for both ResNeXt-50 and ViT-B/16 backbones. Figure 11 summarizes the top-performing single-modality configurations across both encoders.

Across modalities, in-domain performance is relatively similar, but cross-region behavior varies substantially. For ResNeXt-50, EP achieves the highest in-domain scores, but exhibits the largest performance drop under domain shift, whereas S achieves slightly lower peak performance with significantly better transferability. For ViT-B/16, S, DEM, and EP provide the strongest overall results, and cross-region gaps are smaller and more uniform than with ResNeXt-50. These trends indicate that ResNeXt-50 offers higher peak performance, while ViT-B/16 yields more consistent generalization across regions.

## D.2 MULTI-SCALE FUSION

Tables 11, 12, and 13 report complete results for all multi-scale, single-modality experiments for in-domain and cross-domain evaluations of both ResNeXt-50 and ViT-B/16 backbones. Figure 11 summarizes the top-performing models across all multi-scale configurations for both encoders.

Across scales, ResNeXt-50 again achieves the highest peak in-domain performance, with EP leading overall. However, EP experiences the largest cross-region drop, whereas S and SDS retain much more of their performance and exhibit smaller gaps than even in the single-scale setting. For ViT-B/16, S similarly provides the strongest and most stable result, with even smaller cross-region declines than its single-scale counterparts. ViT-B/16 also benefits noticeably from multi-scale cur-

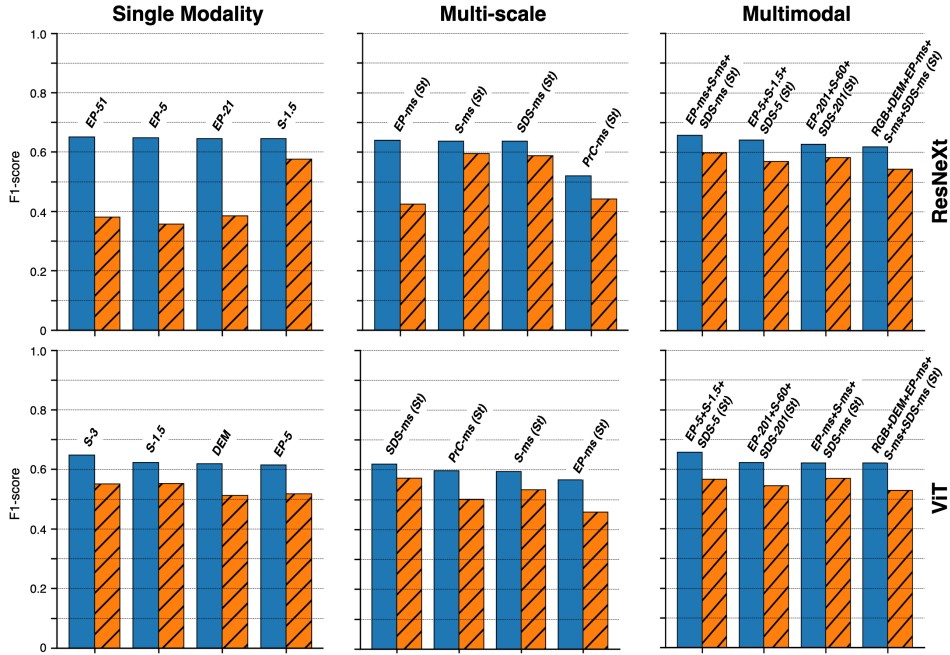

Figure 11: In-domain (blue) and cross-domain (orange, hatched) F1 scores for the top four models for single-modality, multi-scale fusion, and multimodal fusion experiments. Rows show comparisons of ResNeXt-50 (top) vs. ViT-B/16 (bottom) backbones. Each subplot shows the four best-performing models based on in-domain F1 scores. Cross-domain bars illustrate domain shift using the same models selected based on in-domain performance. Model configurations are shown above each group and indicate the input modality, or modality combination and fusion strategy.

vature inputs, with PrC emerging as a relatively strong predictor. Overall, these results indicate that multi-scale terrain derivatives, particularly S and SDS, improve cross-region robustness, and that backbone choice can influence which shape-based cues are most effectively leveraged.

### D.3 MULTIMODAL FUSION

Tables 14, 15, and 16 report complete results for all multimodal fusion experiments for both encoders across in-domain and cross-domain evaluations. These experiments evaluate multiple fusion strategies, including early channel stacking, mid-level concatenation, and mid-level attention variants. Figure 11 summarizes the top-performing multimodal configurations across both encoders.

Across modalities and fusion strategies, early channel stacking consistently performs best. ResNeXt-50 achieves its strongest performance with the multiscale EP+S+SDS combination, which also yields the best cross-region results of any model tested. Multimodal configurations, including those that incorporate RGB or DEM, exhibit relatively small cross-region drops. For ViT-B/16, the highest performance is achieved using single-scale combinations of EP+S+SDS, although cross-region performance is slightly lower than with ResNeXt-50. Overall, multimodal fusion improves robustness for both encoders, with stacking providing the most reliable gains.

### D.4 CLASS-LEVEL TRENDS

Tables Tables 17 and 18 report class-wise AUC for all evaluated models across both in-domain (Warren County) and cross-domain (Hardin County) test sets. Results are provided for all single-modality, multi-scale, and multimodal fusion configurations under both ResNeXt-50 and ViT-B/16 backbones. Figure 12 summarizes the per-class AUC of the top-performing model for each backbone. These results complement the macro-averaged metrics presented earlier in the appendix and provide a detailed view of class-level behavior across modalities, scales, and fusion strategies.

Across encoders and configurations, class-level trends are consistent. ResNeXt-50 performs best on af1, Qal, Qaf, and Qat, whereas ViT-B/16 achieves higher scores on Qc, Qca, and Qr. Multi-scale inputs improve overall performance, but maintain these differences, and multimodal fusion significantly raises class-level scores for ResNeXt-50 while providing more modest gains for ViT-B/16. Performance does not strictly follow class frequency: Qc and Qca perform highest, but have moderate frequency; Qr performs modestly, but is most frequent; Qat and af1 perform modestly, but Qat is a rare class; Qaf also performs relatively well despite its rarity; Qal remains the weakest across all settings, but is the second most common class.

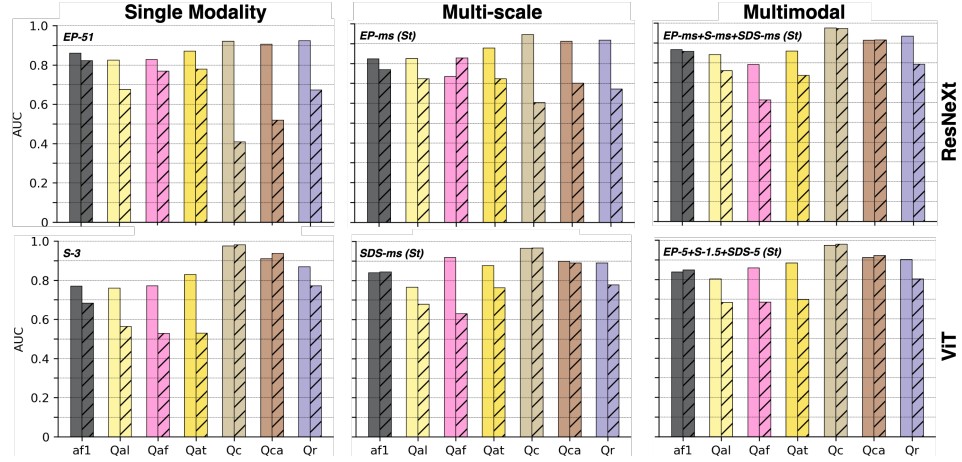

Figure 12: In-domain (solid) and cross-domain (hatched) class-wise AUC scores for the single best-performing models across different experiment types and backbone architectures. Rows show comparisons of ResNeXt-50 (top) vs. ViT-B/16 (bottom) backbones. Each subplot shows the best-performing model based on in-domain F1 scores. Cross-domain bars illustrate domain shift using the same model selected based on in-domain performance. Model configurations are shown above each group and indicate the input modality, or modality combination and fusion strategy.

## D.5 Comparisons with Existing Models

We conducted exploratory experiments with several recent multimodal foundation models, including SatMAE (Cong et al., 2022), SatMAE++ (Noman et al., 2024), DOFA (Xiong et al., 2024), and Panopticon (Waldmann et al., 2025). These models were developed for grouped multispectral or multisensor satellite imagery and are not natively configured to handle LiDAR-derived terrain features at multiple spatial scales. Our goal was not exhaustive hyperparameter optimization, but rather to provide indicative baselines for how existing large-scale models perform on EarthScape. DOFA and Panopticon are both transformer-based foundation models for multimodal Earth observation, and were tested with native inputs of RGB+NIR. Following the grouping strategy of SatMAE and SatMAE++, we organized EarthScape modalities into three groups: (1) RGB+DEM, (2) EP at four scales (1.5, 6, 15, 30 m GSD), and (3) S and SDS at one scale (1.5 m GSD). This configuration included ten modalities drawn from the strongest single-modality performers. Our experiments used the same training, validation, and test splits.

Across all foundation models, in-domain performance was lower than that of SGMap-Net, and cross-region degradation was substantial. SatMAE++ achieved competitive in-domain scores but dropped sharply under domain shift, while DOFA showed relatively small cross-region gaps but had much lower overall accuracy. Panopticon similarly underperformed across both regions. In contrast, the multimodal SGMap-Net variant outperformed all foundation models in both absolute performance and generalization. This indicates that architectures developed for spectral imagery are insufficient for surface-aware tasks, and that a simple, geologically-informed model like SGMap-Net can provide markedly stronger results.

Table 8: Macro-averaged F1 and AUC for *single modality* models on in-domain (ID) and cross-domain (CD) test sets. Results are reported for ResNeXt-50 and ViT-B/16 backbones. ID-CD performance differences ($\Delta$) are also shown. The best and second-best scores in each column are indicated in **bold** and underlined, respectively.

| Modality | F1 (ResNeXt) | | | F1 (ViT) | | | AUC (ResNeXt) | | | AUC (ViT) | | |
|---|---|---|---|---|---|---|---|---|---|---|---|---|
| | ID | CD | $\Delta$ | ID | CD | $\Delta$ | ID | CD | $\Delta$ | ID | CD | $\Delta$ |
| DEM | 0.632 | 0.527 | 0.105 | 0.618 | 0.512 | 0.237 | **0.883** | 0.730 | 0.153 | **0.857** | 0.620 | 0.237 |
| RGB | 0.599 | 0.394 | 0.205 | 0.579 | 0.332 | 0.267 | 0.815 | 0.557 | 0.258 | 0.793 | 0.526 | 0.267 |
| NIR | 0.613 | 0.468 | 0.145 | 0.579 | 0.275 | 0.274 | 0.815 | 0.650 | 0.166 | 0.784 | 0.509 | 0.274 |
| NHD | 0.515 | 0.434 | 0.081 | 0.492 | 0.428 | 0.064 | 0.659 | 0.576 | 0.083 | 0.496 | 0.509 | -0.013 |
| OSM | 0.530 | 0.463 | 0.067 | 0.500 | 0.428 | 0.072 | 0.653 | 0.587 | 0.066 | 0.545 | 0.513 | 0.032 |
| $EP_5$ | 0.648 | 0.357 | 0.291 | 0.614 | 0.518 | 0.117 | 0.872 | 0.582 | 0.290 | 0.854 | 0.738 | 0.117 |
| $EP_{11}$ | 0.639 | 0.425 | 0.214 | 0.603 | 0.519 | 0.082 | 0.879 | 0.675 | 0.203 | 0.850 | **0.768** | 0.082 |
| $EP_{21}$ | 0.645 | 0.384 | 0.261 | 0.608 | 0.503 | 0.079 | 0.877 | 0.695 | 0.183 | 0.838 | 0.759 | 0.079 |
| $EP_{51}$ | **0.651** | 0.380 | 0.271 | 0.604 | 0.489 | 0.078 | 0.876 | 0.663 | 0.213 | 0.835 | 0.757 | 0.078 |
| $EP_{101}$ | 0.619 | 0.476 | 0.143 | 0.589 | 0.477 | 0.075 | 0.857 | 0.739 | 0.118 | 0.819 | 0.744 | 0.075 |
| $EP_{201}$ | 0.610 | 0.391 | 0.219 | 0.584 | 0.472 | 0.062 | 0.869 | 0.724 | 0.145 | 0.799 | 0.737 | 0.062 |
| $PlC_{1.5}$ | 0.491 | 0.425 | 0.066 | 0.517 | 0.452 | 0.013 | 0.514 | 0.513 | **0.001** | 0.603 | 0.590 | 0.013 |
| $PlC_3$ | 0.494 | 0.426 | 0.068 | 0.524 | 0.457 | 0.007 | 0.501 | 0.500 | **0.001** | 0.621 | 0.614 | 0.007 |
| $PlC_6$ | 0.495 | 0.425 | 0.070 | 0.513 | 0.453 | **0.005** | 0.488 | 0.485 | 0.002 | 0.632 | 0.627 | **0.005** |
| $PlC_{15}$ | 0.488 | 0.425 | 0.063 | 0.495 | 0.426 | 0.016 | 0.472 | 0.459 | 0.013 | 0.560 | 0.544 | 0.016 |
| $PlC_{30}$ | 0.488 | 0.420 | 0.068 | 0.484 | 0.422 | -0.008 | 0.511 | 0.470 | 0.041 | 0.532 | 0.540 | -0.008 |
| $PlC_{60}$ | 0.488 | 0.433 | 0.055 | 0.495 | 0.427 | -0.039 | 0.474 | 0.528 | -0.054 | 0.500 | 0.539 | -0.039 |
| $PrC_{1.5}$ | 0.493 | 0.433 | 0.060 | 0.494 | 0.426 | -0.039 | 0.554 | 0.516 | 0.038 | 0.407 | 0.446 | -0.039 |
| $PrC_3$ | 0.492 | 0.421 | 0.071 | 0.497 | 0.425 | 0.023 | 0.486 | 0.520 | -0.034 | 0.517 | 0.493 | 0.023 |
| $PrC_6$ | 0.496 | 0.415 | 0.081 | 0.495 | 0.426 | -0.055 | 0.508 | 0.463 | 0.046 | 0.389 | 0.444 | -0.055 |
| $PrC_{15}$ | 0.492 | 0.417 | 0.074 | 0.494 | 0.426 | -0.022 | 0.440 | 0.398 | 0.042 | 0.466 | 0.487 | -0.022 |
| $PrC_{30}$ | 0.510 | 0.418 | 0.092 | 0.540 | 0.431 | 0.035 | 0.553 | 0.491 | 0.062 | 0.613 | 0.578 | 0.035 |
| $PrC_{60}$ | 0.495 | 0.425 | 0.071 | 0.549 | 0.431 | 0.028 | 0.417 | 0.428 | -0.011 | 0.626 | 0.599 | 0.028 |
| $S_{1.5}$ | 0.645 | **0.575** | 0.070 | 0.623 | 0.552 | 0.093 | 0.876 | **0.808** | 0.068 | 0.855 | 0.762 | 0.093 |
| $S_3$ | 0.619 | 0.570 | 0.049 | **0.647** | 0.551 | 0.127 | 0.875 | 0.779 | 0.096 | 0.841 | 0.713 | 0.127 |
| $S_6$ | 0.617 | 0.555 | 0.061 | 0.614 | **0.555** | 0.102 | 0.861 | 0.804 | 0.057 | 0.833 | 0.731 | 0.102 |
| $S_{15}$ | 0.612 | 0.537 | 0.075 | 0.600 | 0.554 | 0.081 | 0.841 | 0.744 | 0.096 | 0.812 | 0.731 | 0.081 |
| $S_{30}$ | 0.594 | 0.536 | 0.058 | 0.578 | 0.528 | 0.061 | 0.811 | 0.710 | 0.102 | 0.765 | 0.705 | 0.061 |
| $S_{60}$ | 0.543 | 0.485 | 0.058 | 0.578 | 0.514 | 0.093 | 0.601 | 0.578 | 0.023 | 0.770 | 0.676 | 0.093 |
| $SDS_5$ | 0.613 | 0.567 | 0.045 | 0.569 | 0.513 | 0.072 | 0.850 | 0.804 | 0.046 | 0.786 | 0.713 | 0.072 |
| $SDS_{11}$ | 0.631 | **0.575** | 0.056 | 0.599 | 0.543 | 0.080 | 0.846 | 0.786 | 0.061 | 0.803 | 0.723 | 0.080 |
| $SDS_{21}$ | 0.633 | 0.573 | 0.060 | 0.591 | 0.552 | 0.074 | 0.854 | 0.786 | 0.067 | 0.809 | 0.735 | 0.074 |
| $SDS_{51}$ | 0.603 | 0.533 | 0.069 | 0.554 | 0.536 | 0.038 | 0.841 | 0.746 | 0.095 | 0.727 | 0.689 | 0.038 |
| $SDS_{101}$ | 0.611 | 0.571 | **0.040** | 0.535 | 0.502 | 0.037 | 0.848 | 0.756 | 0.092 | 0.718 | 0.681 | 0.037 |
| $SDS_{201}$ | 0.613 | 0.527 | 0.086 | 0.548 | 0.508 | 0.064 | 0.837 | 0.713 | 0.124 | 0.735 | 0.671 | 0.064 |

Table 9: Macro-averaged precision and recall for _single modality_ models on in-domain (ID) and cross-domain (CD) test sets. Results are reported for ResNeXt-50 and ViT-B/16 backbones. ID-CD performance differences ($\Delta$) are also shown. The best and second-best scores in each column are indicated in **bold** and underlined, respectively.

| Modality | Precision (ResNeXt) | | | Precision (ViT) | | | Recall (ResNeXt) | | | Recall (ViT) | | |
|---|---|---|---|---|---|---|---|---|---|---|---|---|
| | ID | CD | $\Delta$ | ID | CD | $\Delta$ | ID | CD | $\Delta$ | ID | CD | $\Delta$ |
| DEM | 0.621 | 0.460 | 0.161 | 0.551 | 0.432 | 0.125 | 0.661 | 0.653 | 0.008 | 0.800 | 0.674 | 0.125 |
| RGB | 0.553 | 0.405 | 0.148 | 0.522 | 0.296 | 0.235 | 0.672 | 0.418 | 0.254 | 0.664 | 0.429 | 0.235 |
| NIR | 0.564 | 0.486 | 0.078 | 0.521 | 0.273 | 0.384 | 0.698 | 0.514 | 0.184 | 0.668 | 0.284 | 0.384 |
| NHD | 0.419 | 0.353 | 0.066 | 0.390 | 0.334 | 0.056 | 0.725 | 0.691 | 0.034 | 0.857 | 0.881 | -0.024 |
| OSM | 0.442 | 0.373 | 0.069 | 0.395 | 0.334 | 0.061 | 0.846 | 0.853 | -0.007 | 0.971 | 0.949 | 0.022 |
| EP$_5$ | 0.617 | 0.450 | 0.167 | 0.556 | 0.452 | 0.112 | 0.706 | 0.333 | 0.373 | 0.733 | 0.621 | 0.112 |
| EP$_{11}$ | 0.602 | 0.474 | 0.128 | 0.552 | 0.449 | 0.060 | 0.748 | 0.428 | 0.320 | 0.690 | 0.631 | 0.060 |
| EP$_{21}$ | **0.629** | 0.455 | 0.173 | 0.548 | 0.435 | 0.089 | 0.737 | 0.416 | 0.321 | 0.706 | 0.617 | 0.089 |
| EP$_{51}$ | 0.612 | 0.382 | 0.230 | 0.565 | 0.440 | 0.087 | 0.705 | 0.389 | 0.316 | 0.664 | 0.577 | 0.087 |
| EP$_{101}$ | 0.570 | 0.480 | 0.090 | 0.539 | 0.421 | 0.102 | 0.727 | 0.551 | 0.176 | 0.674 | 0.572 | 0.102 |
| EP$_{201}$ | 0.593 | 0.465 | 0.127 | 0.520 | 0.425 | 0.092 | 0.634 | 0.364 | 0.270 | 0.707 | 0.615 | 0.092 |
| PlC$_{1.5}$ | 0.390 | 0.333 | 0.057 | 0.419 | 0.359 | 0.078 | 0.837 | 0.829 | 0.007 | 0.806 | 0.728 | 0.078 |
| PlC$_3$ | 0.391 | 0.333 | 0.059 | 0.432 | 0.370 | 0.119 | **1.000** | **1.000** | **0.000** | 0.871 | 0.752 | 0.119 |
| PlC$_6$ | 0.393 | 0.333 | 0.060 | 0.429 | 0.365 | 0.052 | 0.892 | 0.889 | 0.003 | 0.853 | 0.801 | 0.052 |
| PlC$_{30}$ | 0.390 | 0.332 | 0.058 | 0.392 | 0.334 | -0.045 | 0.856 | 0.809 | 0.047 | 0.795 | 0.840 | -0.045 |
| PlC$_{15}$ | 0.390 | 0.334 | 0.057 | 0.403 | 0.338 | -0.029 | 0.823 | 0.834 | -0.010 | 0.765 | 0.794 | -0.029 |
| PlC$_{60}$ | 0.389 | 0.337 | **0.052** | 0.393 | 0.335 | -0.022 | 0.842 | 0.921 | -0.079 | 0.973 | 0.995 | -0.022 |
| PrC$_{1.5}$ | 0.392 | 0.341 | **0.052** | 0.391 | 0.333 | **0.000** | 0.967 | 0.946 | 0.021 | **1.000** | **1.000** | **0.000** |
| PrC$_3$ | 0.394 | 0.335 | 0.059 | 0.406 | 0.336 | **0.000** | 0.819 | 0.853 | -0.034 | 0.919 | 0.919 | **0.000** |
| PrC$_6$ | 0.396 | 0.328 | 0.068 | 0.392 | 0.333 | -0.001 | 0.739 | 0.719 | 0.020 | 0.997 | 0.998 | -0.001 |
| PrC$_{15}$ | 0.392 | 0.331 | 0.061 | 0.391 | 0.333 | **0.000** | 0.759 | 0.718 | 0.041 | **1.000** | **1.000** | **0.000** |
| PrC$_{30}$ | 0.430 | 0.337 | 0.092 | 0.456 | 0.348 | 0.074 | 0.679 | 0.639 | 0.040 | 0.731 | 0.657 | 0.074 |
| PrC$_{60}$ | 0.392 | 0.332 | 0.060 | 0.464 | 0.350 | 0.100 | 0.896 | 0.854 | 0.042 | 0.748 | 0.648 | 0.100 |
| S$_{1.5}$ | 0.616 | 0.506 | 0.110 | 0.578 | 0.489 | 0.051 | 0.681 | 0.687 | -0.006 | 0.726 | 0.674 | 0.051 |
| S$_3$ | 0.590 | **0.507** | 0.084 | **0.614** | 0.490 | 0.041 | 0.654 | 0.662 | -0.009 | 0.693 | 0.653 | 0.041 |
| S$_6$ | 0.592 | 0.497 | 0.095 | 0.553 | **0.491** | 0.072 | 0.670 | 0.671 | 0.001 | 0.791 | 0.720 | 0.072 |
| S$_{15}$ | 0.550 | 0.478 | 0.072 | 0.537 | 0.484 | -0.027 | 0.749 | 0.664 | 0.085 | 0.774 | 0.801 | -0.027 |
| S$_{30}$ | 0.523 | 0.464 | 0.059 | 0.508 | 0.464 | 0.054 | 0.744 | 0.679 | 0.065 | 0.717 | 0.663 | 0.054 |
| S$_{60}$ | 0.469 | 0.409 | 0.060 | 0.500 | 0.436 | 0.064 | 0.697 | 0.651 | 0.047 | 0.736 | 0.672 | 0.064 |
| SDS$_5$ | 0.580 | 0.487 | 0.093 | 0.518 | 0.435 | -0.025 | 0.661 | 0.707 | -0.047 | 0.641 | 0.666 | -0.025 |
| SDS$_{11}$ | 0.596 | 0.499 | 0.097 | 0.545 | 0.460 | 0.084 | 0.689 | 0.698 | -0.008 | 0.769 | 0.685 | 0.084 |
| SDS$_{21}$ | 0.578 | 0.486 | 0.092 | 0.529 | 0.469 | -0.006 | 0.768 | 0.740 | 0.027 | 0.690 | 0.696 | -0.006 |
| SDS$_{51}$ | 0.578 | 0.471 | 0.108 | 0.482 | 0.443 | 0.022 | 0.638 | 0.646 | -0.008 | 0.740 | 0.718 | 0.022 |
| SDS$_{101}$ | 0.566 | 0.490 | 0.075 | 0.459 | 0.409 | -0.009 | 0.775 | 0.716 | 0.058 | 0.710 | 0.719 | -0.009 |
| SDS$_{201}$ | 0.558 | 0.452 | 0.107 | 0.459 | 0.411 | 0.044 | 0.709 | 0.660 | 0.048 | 0.796 | 0.752 | 0.044 |

Table 10: Mean average precision (mAP) and macro-averaged accuracy for *single modality* models on in-domain (ID) and cross-domain (CD) test sets. Results are reported for ResNeXt-50 and ViT-B/16 backbones. ID-CD performance differences ($\Delta$) are also shown. The best and second-best scores in each column are indicated in **bold** and underlined, respectively.

| Modality | mAP (ResNeXt) | | | mAP (ViT) | | | Accuracy (ResNeXt) | | | Accuracy (ViT) | | |
|---|---|---|---|---|---|---|---|---|---|---|---|---|
| | ID | CD | $\Delta$ | ID | CD | $\Delta$ | ID | CD | $\Delta$ | ID | CD | $\Delta$ |
| DEM | 0.554 | 0.442 | 0.111 | 0.516 | 0.431 | 0.022 | **0.873** | 0.827 | 0.046 | 0.808 | 0.785 | 0.022 |
| RGB | 0.509 | 0.367 | 0.143 | 0.489 | 0.336 | 0.109 | 0.832 | 0.781 | 0.051 | 0.815 | 0.706 | 0.109 |
| NIR | 0.513 | 0.387 | 0.125 | 0.485 | 0.337 | 0.020 | 0.833 | 0.809 | 0.025 | 0.812 | 0.792 | 0.020 |
| NHD | 0.403 | 0.339 | 0.064 | 0.391 | 0.333 | 0.058 | 0.682 | 0.634 | 0.048 | 0.523 | 0.468 | 0.055 |
| OSM | 0.435 | 0.367 | 0.068 | 0.395 | 0.334 | 0.061 | 0.647 | 0.548 | 0.099 | 0.545 | 0.406 | 0.139 |
| $EP_5$ | 0.549 | 0.385 | 0.164 | 0.516 | 0.417 | 0.019 | 0.858 | 0.831 | 0.026 | 0.829 | 0.810 | 0.019 |
| $EP_{11}$ | 0.551 | 0.397 | 0.154 | 0.510 | 0.409 | 0.024 | 0.854 | 0.832 | 0.022 | 0.829 | 0.805 | 0.024 |
| $EP_{21}$ | **0.565** | 0.386 | 0.179 | 0.504 | 0.398 | 0.029 | 0.860 | 0.828 | 0.031 | 0.827 | 0.798 | 0.029 |
| $EP_{51}$ | 0.546 | 0.377 | 0.169 | 0.507 | 0.395 | 0.034 | 0.862 | 0.818 | 0.044 | 0.837 | 0.803 | 0.034 |
| $EP_{101}$ | 0.528 | 0.401 | 0.128 | 0.500 | 0.385 | 0.034 | 0.835 | 0.812 | 0.024 | 0.818 | 0.784 | 0.034 |
| $EP_{201}$ | 0.535 | 0.381 | 0.154 | 0.476 | 0.367 | 0.041 | 0.858 | 0.838 | 0.019 | 0.791 | 0.750 | 0.041 |
| $PlC_{1.5}$ | 0.391 | 0.333 | 0.058 | 0.411 | 0.354 | 0.015 | 0.551 | 0.502 | 0.049 | 0.643 | 0.628 | 0.015 |
| $PlC_3$ | 0.391 | 0.333 | 0.059 | 0.418 | 0.353 | 0.005 | 0.392 | 0.333 | 0.059 | 0.631 | 0.626 | 0.005 |
| $PlC_6$ | 0.393 | 0.333 | 0.060 | 0.416 | 0.353 | **-0.001** | 0.494 | 0.452 | 0.043 | 0.617 | 0.619 | **-0.001** |
| $PlC_{15}$ | 0.391 | 0.334 | 0.057 | 0.397 | 0.335 | 0.053 | 0.533 | 0.482 | 0.051 | 0.644 | 0.591 | 0.053 |
| $PlC_{30}$ | 0.392 | 0.333 | 0.059 | 0.392 | 0.334 | 0.064 | 0.524 | 0.467 | 0.057 | 0.586 | 0.521 | 0.064 |
| $PlC_{60}$ | 0.390 | 0.335 | 0.055 | 0.393 | 0.335 | 0.062 | 0.525 | 0.471 | 0.054 | 0.456 | 0.395 | 0.062 |
| $PrC_{1.5}$ | 0.392 | 0.340 | 0.052 | 0.391 | 0.333 | 0.059 | 0.411 | 0.402 | **0.009** | 0.392 | 0.333 | 0.059 |
| $PrC_3$ | 0.393 | 0.332 | 0.060 | 0.400 | 0.334 | 0.051 | 0.527 | 0.466 | 0.061 | 0.452 | 0.401 | 0.051 |
| $PrC_6$ | 0.392 | 0.333 | 0.059 | 0.392 | 0.333 | 0.062 | 0.645 | 0.581 | 0.064 | 0.395 | 0.334 | 0.062 |
| $PrC_{15}$ | 0.393 | 0.334 | 0.059 | 0.391 | 0.333 | 0.059 | 0.644 | 0.591 | 0.054 | 0.392 | 0.333 | 0.059 |
| $PrC_{30}$ | 0.406 | 0.339 | 0.067 | 0.431 | 0.345 | 0.055 | 0.714 | 0.674 | 0.040 | 0.726 | 0.671 | 0.055 |
| $PrC_{60}$ | 0.392 | 0.333 | 0.059 | 0.433 | 0.345 | 0.045 | 0.510 | 0.463 | 0.047 | 0.723 | 0.677 | 0.045 |
| $S_{1.5}$ | 0.552 | 0.468 | 0.084 | 0.525 | 0.456 | 0.021 | 0.871 | 0.848 | 0.023 | 0.840 | 0.819 | 0.021 |
| $S_3$ | 0.543 | **0.472** | 0.071 | **0.542** | 0.465 | 0.025 | 0.867 | **0.852** | 0.015 | **0.850** | **0.825** | 0.025 |
| $S_6$ | 0.539 | 0.463 | 0.077 | 0.523 | **0.466** | 0.019 | 0.857 | 0.844 | 0.013 | 0.812 | 0.793 | 0.019 |
| $S_{15}$ | 0.517 | 0.455 | 0.062 | 0.506 | 0.463 | 0.012 | 0.807 | 0.799 | 0.008 | 0.794 | 0.781 | 0.012 |
| $S_{30}$ | 0.501 | 0.447 | 0.053 | 0.485 | 0.452 | **-0.001** | 0.793 | 0.784 | 0.009 | 0.792 | 0.793 | **-0.001** |
| $S_{60}$ | 0.450 | 0.398 | 0.052 | 0.481 | 0.435 | 0.003 | 0.742 | 0.752 | -0.010 | 0.784 | 0.780 | 0.003 |
| $SDS_5$ | 0.527 | 0.459 | 0.068 | 0.484 | 0.420 | 0.011 | 0.853 | 0.833 | 0.020 | 0.820 | 0.809 | 0.011 |
| $SDS_{11}$ | 0.533 | 0.466 | 0.068 | 0.504 | 0.434 | 0.011 | 0.850 | 0.839 | 0.011 | 0.806 | 0.795 | 0.011 |
| $SDS_{21}$ | 0.531 | 0.454 | 0.078 | 0.491 | 0.435 | 0.007 | 0.836 | 0.819 | 0.017 | 0.816 | 0.809 | 0.007 |
| $SDS_{51}$ | 0.529 | 0.436 | 0.093 | 0.459 | 0.418 | 0.002 | 0.855 | 0.824 | 0.031 | 0.754 | 0.752 | 0.002 |
| $SDS_{101}$ | 0.525 | 0.461 | 0.064 | 0.448 | 0.400 | -0.017 | 0.820 | 0.808 | 0.012 | 0.734 | 0.751 | -0.017 |
| $SDS_{201}$ | 0.520 | 0.427 | 0.093 | 0.446 | 0.402 | -0.019 | 0.834 | 0.805 | 0.030 | 0.710 | 0.729 | -0.019 |

Table 11: Macro-averaged F1 and AUC for *multi-scale fusion* models on in-domain (ID) and cross-domain (CD) test sets. Results are reported for ResNeXt-50 and ViT-B/16 backbones under two fusion strategies: early channel stacking (St) and cross-attention with a shared encoder (A1). ID–CD performance differences ($\Delta$) are also shown. The best and second-best scores in each column are indicated in **bold** and underlined, respectively.

| Modality / Fusion | F1 (ResNeXt) | | | F1 (ViT) | | | AUC (ResNeXt) | | | AUC (ViT) | | |
|---|---|---|---|---|---|---|---|---|---|---|---|---|
| | ID | CD | $\Delta$ | ID | CD | $\Delta$ | ID | CD | $\Delta$ | ID | CD | $\Delta$ |
| $EP_{ms}$ (St) | **0.640** | 0.425 | 0.215 | 0.566 | 0.458 | 0.108 | 0.862 | 0.717 | 0.145 | 0.756 | 0.693 | 0.063 |
| $PlC_{ms}$ (St) | 0.490 | 0.426 | 0.063 | 0.493 | 0.429 | 0.063 | 0.525 | 0.521 | 0.004 | 0.511 | 0.536 | -0.026 |
| $PrC_{ms}$ (St) | 0.519 | 0.441 | 0.078 | 0.596 | 0.501 | 0.095 | 0.579 | 0.497 | 0.082 | **0.816** | **0.727** | 0.089 |
| $S_{ms}$ (St) | 0.637 | **0.594** | 0.043 | 0.593 | 0.533 | 0.061 | 0.864 | **0.804** | 0.061 | 0.798 | 0.705 | 0.093 |
| $SDS_{ms}$ (St) | 0.636 | 0.588 | 0.048 | **0.619** | **0.571** | 0.048 | **0.878** | 0.792 | 0.086 | 0.672 | 0.644 | 0.028 |
| $EP_{ms}$ (A1) | 0.494 | 0.426 | 0.068 | 0.561 | 0.445 | 0.117 | 0.500 | 0.500 | **0.000** | 0.759 | 0.664 | 0.095 |
| $PlC_{ms}$ (A1) | 0.494 | 0.426 | 0.068 | 0.505 | 0.435 | 0.070 | 0.500 | 0.500 | **0.000** | 0.578 | 0.581 | -0.003 |
| $PrC_{ms}$ (A1) | 0.494 | 0.426 | 0.068 | 0.531 | 0.410 | 0.121 | 0.500 | 0.500 | **0.000** | 0.594 | 0.562 | 0.032 |
| $S_{ms}$ (A1) | 0.494 | 0.426 | 0.068 | 0.557 | 0.519 | **0.038** | 0.500 | 0.500 | **0.000** | 0.615 | 0.594 | 0.021 |
| $SDS_{ms}$ (A1) | 0.493 | 0.451 | **0.042** | 0.494 | 0.426 | 0.068 | 0.618 | 0.618 | 0.001 | 0.500 | 0.500 | **0.000** |

Table 12: Macro-averaged precision and recall for *multi-scale fusion* models on in-domain (ID) and cross-domain (CD) test sets. Results are reported for ResNeXt-50 and ViT-B/16 backbones under two fusion strategies: early channel stacking (St) and cross-attention with a shared encoder (A1). ID–CD performance differences ($\Delta$) are also shown. The best and second-best scores in each column are indicated in **bold** and underlined, respectively.

| Modality / Fusion | Precision (ResNeXt) | | | Precision (ViT) | | | Recall (ResNeXt) | | | Recall (ViT) | | |
|---|---|---|---|---|---|---|---|---|---|---|---|---|
| | ID | CD | $\Delta$ | ID | CD | $\Delta$ | ID | CD | $\Delta$ | ID | CD | $\Delta$ |
| $EP_{ms}$ (St) | 0.606 | **0.556** | **0.051** | 0.493 | 0.380 | 0.112 | 0.703 | 0.426 | 0.277 | 0.712 | 0.636 | 0.076 |
| $PlC_{ms}$ (St) | 0.391 | 0.335 | 0.056 | 0.391 | 0.335 | 0.056 | 0.738 | 0.738 | 0.000 | 0.872 | 0.940 | -0.067 |
| $PrC_{ms}$ (St) | 0.429 | 0.353 | 0.076 | 0.530 | 0.435 | 0.095 | 0.697 | 0.694 | 0.003 | 0.743 | 0.642 | 0.101 |
| $S_{ms}$ (St) | **0.607** | 0.535 | 0.072 | 0.525 | 0.455 | 0.070 | 0.730 | 0.682 | 0.047 | 0.714 | 0.681 | 0.033 |
| $SDS_{ms}$ (St) | 0.588 | 0.509 | 0.079 | **0.575** | **0.472** | 0.103 | 0.742 | 0.729 | 0.013 | 0.675 | 0.674 | 0.001 |
| $EP_{ms}$ (A1) | 0.391 | 0.333 | 0.059 | 0.483 | 0.375 | 0.108 | **1.000** | **1.000** | 0.000 | 0.700 | 0.612 | 0.088 |
| $PlC_{ms}$ (A1) | 0.391 | 0.333 | 0.059 | 0.405 | 0.341 | 0.064 | **1.000** | **1.000** | 0.000 | 0.874 | 0.868 | 0.006 |
| $PrC_{ms}$ (A1) | 0.391 | 0.333 | 0.059 | 0.431 | 0.325 | 0.106 | **1.000** | **1.000** | 0.000 | 0.738 | 0.678 | 0.060 |
| $S_{ms}$ (A1) | 0.391 | 0.333 | 0.058 | 0.489 | 0.440 | **0.049** | **1.000** | **1.000** | 0.000 | 0.745 | 0.688 | 0.057 |
| $SDS_{ms}$ (A1) | 0.432 | 0.380 | 0.052 | 0.391 | 0.332 | 0.057 | 0.801 | 0.748 | 0.053 | **1.000** | **1.000** | **0.000** |

Table 13: Mean average precision (mAP) and macro-averaged accuracy for *multi-scale fusion* models on in-domain (ID) and cross-domain (CD) test sets. Results are reported for ResNeXt-50 and ViT-B/16 backbones under two fusion strategies: early channel stacking (St) and cross-attention with a shared encoder (A1). ID–CD performance differences ($\Delta$) are also shown. The best and second-best scores in each column are indicated in **bold** and underlined, respectively.

| Modality / Fusion | mAP (ResNeXt) | | | mAP (ViT) | | | Accuracy (ResNeXt) | | | Accuracy (ViT) | | |
|---|---|---|---|---|---|---|---|---|---|---|---|---|
| | ID | CD | $\Delta$ | ID | CD | $\Delta$ | ID | CD | $\Delta$ | ID | CD | $\Delta$ |
| $EP_{ms}$ (St) | 0.555 | 0.403 | 0.152 | 0.460 | 0.360 | 0.099 | **0.865** | 0.828 | 0.037 | 0.774 | 0.724 | 0.050 |
| $PlC_{ms}$ (St) | 0.392 | 0.335 | **0.057** | 0.392 | 0.335 | 0.057 | 0.634 | 0.588 | 0.046 | 0.534 | 0.465 | 0.069 |
| $PrC_{ms}$ (St) | 0.416 | 0.348 | 0.069 | 0.504 | 0.423 | 0.081 | 0.717 | 0.666 | 0.051 | 0.794 | 0.768 | 0.027 |
| $S_{ms}$ (St) | **0.557** | **0.491** | 0.066 | 0.498 | **0.453** | 0.045 | 0.856 | **0.860** | **-0.004** | 0.810 | 0.803 | 0.006 |
| $SDS_{ms}$ (St) | 0.540 | 0.470 | 0.070 | **0.522** | 0.447 | 0.075 | 0.846 | 0.839 | 0.007 | **0.851** | **0.826** | 0.025 |
| $EP_{ms}$ (A1) | 0.391 | 0.333 | 0.059 | 0.450 | 0.362 | 0.088 | 0.391 | 0.333 | 0.059 | 0.766 | 0.727 | 0.039 |
| $PlC_{ms}$ (A1) | 0.391 | 0.333 | 0.059 | 0.401 | 0.338 | 0.062 | 0.391 | 0.333 | 0.059 | 0.598 | 0.541 | 0.057 |
| $PrC_{ms}$ (A1) | 0.391 | 0.333 | 0.059 | 0.407 | 0.333 | 0.074 | 0.391 | 0.333 | 0.059 | 0.691 | 0.625 | 0.065 |
| $S_{ms}$ (A1) | 0.391 | 0.333 | 0.058 | 0.472 | 0.434 | **0.038** | 0.391 | 0.333 | 0.059 | 0.742 | 0.747 | **-0.005** |
| $SDS_{ms}$ (A1) | 0.416 | 0.357 | 0.059 | 0.391 | 0.333 | 0.058 | 0.630 | 0.666 | -0.036 | 0.391 | 0.333 | 0.058 |

Table 14: Macro-averaged F1 and AUC for _multimodal fusion_ models on in-domain (ID) and cross-domain (CD) test sets. Results are reported for ResNeXt-50 and ViT-B/16 backbones under four fusion strategies: early channel stacking (St), concatenation of modality embeddings (C), cross-attention with a shared encoder (A1), and cross-attention with separate encoders (A2). ID–CD performance differences ($\Delta$) are also shown. The best and second-best scores in each column are indicated in **bold** and underlined, respectively.

| Modality / Fusion | F1 (ResNeXt) | | | F1 (ViT) | | | AUC (ResNeXt) | | | AUC (ViT) | | |
|---|---|---|---|---|---|---|---|---|---|---|---|---|
| | ID | CD | $\Delta$ | ID | CD | $\Delta$ | ID | CD | $\Delta$ | ID | CD | $\Delta$ |
| $EP_{ms}+S_{ms}+SDS_{ms}$ (St) | **0.657** | **0.598** | 0.059 | 0.621 | **0.569** | 0.053 | 0.882 | 0.806 | 0.076 | 0.860 | **0.774** | 0.086 |
| $EP_5+S_{1.5}+SDS_5$ (St) | 0.641 | 0.568 | 0.073 | **0.657** | 0.566 | 0.092 | 0.848 | **0.812** | 0.036 | 0.712 | 0.664 | 0.048 |
| $EP_{201}+S_{60}+SDS_{201}$ (St) | 0.626 | 0.582 | 0.045 | 0.622 | 0.544 | 0.078 | **0.885** | 0.812 | 0.073 | 0.695 | 0.631 | 0.064 |
| $EP_{ms}+S_{ms}+SDS_{ms}$ (C) | 0.596 | 0.569 | **0.028** | 0.613 | 0.532 | 0.081 | 0.829 | 0.750 | 0.079 | 0.686 | 0.622 | 0.064 |
| RGB+DEM (C) | 0.600 | 0.389 | 0.211 | 0.614 | 0.503 | 0.111 | 0.808 | 0.535 | 0.273 | **0.870** | 0.721 | 0.149 |
| RGB+DEM+$EP_{ms}+S_{ms}+SDS_{ms}$ (C) | 0.618 | 0.543 | 0.074 | 0.621 | 0.528 | 0.093 | 0.858 | 0.739 | 0.118 | 0.735 | 0.615 | 0.120 |
| $EP_{ms}+S_{ms}+SDS_{ms}$ (A1) | 0.561 | 0.532 | 0.029 | 0.567 | 0.538 | **0.029** | 0.677 | 0.707 | -0.030 | 0.776 | 0.678 | 0.098 |
| RGB+DEM (A1) | 0.551 | 0.457 | 0.094 | 0.575 | 0.404 | 0.171 | 0.714 | 0.552 | 0.163 | 0.787 | 0.622 | 0.165 |
| $EP_{ms}+S_{ms}+SDS_{ms}$ (A2) | 0.561 | 0.532 | 0.029 | 0.496 | 0.425 | 0.071 | 0.677 | 0.707 | -0.030 | 0.523 | 0.480 | **0.043** |
| RGB+DEM (A2) | 0.559 | 0.474 | 0.085 | 0.581 | 0.464 | 0.118 | 0.763 | 0.641 | 0.122 | 0.810 | 0.724 | 0.085 |
| RGB+DEM+$EP_{ms}+S_{ms}+SDS_{ms}$ (A2) | 0.494 | 0.426 | 0.068 | 0.520 | 0.457 | 0.063 | 0.500 | 0.500 | **0.000** | 0.572 | 0.511 | 0.061 |

Table 15: Macro-averaged precision and recall for _multimodal fusion_ models on in-domain (ID) and cross-domain (CD) test sets. Results are reported for ResNeXt-50 and ViT-B/16 backbones under four fusion strategies: early channel stacking (St), concatenation of modality embeddings (C), cross-attention with a shared encoder (A1), and cross-attention with separate encoders (A2). ID–CD performance differences ($\Delta$) are also shown. The best and second-best scores in each column are indicated in **bold** and underlined, respectively.

| Modality / Fusion | Precision (ResNeXt) | | | Precision (ViT) | | | Recall (ResNeXt) | | | Recall (ViT) | | |
|---|---|---|---|---|---|---|---|---|---|---|---|---|
| | ID | CD | $\Delta$ | ID | CD | $\Delta$ | ID | CD | $\Delta$ | ID | CD | $\Delta$ |
| $EP_{ms}+S_{ms}+SDS_{ms}$ (St) | **0.626** | **0.546** | 0.080 | 0.568 | 0.491 | 0.077 | 0.735 | 0.666 | 0.068 | 0.761 | 0.711 | 0.050 |
| $EP_5+S_{1.5}+SDS_5$ (St) | 0.606 | 0.531 | 0.074 | 0.604 | 0.482 | 0.122 | 0.697 | 0.623 | 0.074 | 0.731 | 0.708 | 0.023 |
| $EP_{201}+S_{60}+SDS_{201}$ (St) | 0.588 | 0.529 | 0.059 | 0.579 | 0.499 | 0.080 | 0.721 | 0.674 | 0.048 | 0.686 | 0.610 | 0.076 |
| $EP_{ms}+S_{ms}+SDS_{ms}$ (C) | 0.542 | 0.529 | **0.013** | 0.541 | 0.456 | 0.085 | 0.694 | 0.640 | 0.054 | 0.752 | 0.671 | 0.081 |
| RGB+DEM (C) | 0.537 | 0.373 | 0.163 | 0.558 | 0.420 | 0.137 | 0.715 | 0.437 | 0.278 | 0.706 | 0.661 | 0.045 |
| RGB+DEM+$EP_{ms}+S_{ms}+SDS_{ms}$ (C) | 0.563 | 0.496 | 0.067 | 0.574 | 0.485 | 0.090 | 0.740 | 0.644 | 0.096 | 0.621 | 0.622 | **-0.001** |
| $EP_{ms}+S_{ms}+SDS_{ms}$ (A1) | 0.487 | 0.451 | 0.036 | 0.507 | 0.466 | 0.041 | 0.734 | 0.723 | 0.011 | 0.752 | 0.693 | 0.059 |
| RGB+DEM (A1) | 0.495 | 0.445 | 0.050 | 0.515 | 0.387 | 0.129 | 0.647 | 0.555 | 0.092 | 0.686 | 0.582 | 0.105 |
| $EP_{ms}+S_{ms}+SDS_{ms}$ (A2) | 0.487 | 0.451 | 0.036 | 0.392 | 0.332 | 0.060 | 0.734 | 0.723 | 0.011 | **0.984** | **0.889** | 0.095 |
| RGB+DEM (A2) | 0.498 | 0.411 | 0.087 | 0.513 | 0.434 | 0.079 | 0.656 | 0.595 | 0.061 | 0.720 | 0.607 | 0.113 |
| RGB+DEM+$EP_{ms}+S_{ms}+SDS_{ms}$ (A2) | 0.391 | 0.333 | 0.059 | 0.448 | 0.420 | **0.028** | 1.000 | 1.000 | **0.000** | 0.873 | 0.689 | 0.184 |

Table 16: Mean average precision (mAP) and macro-averaged accuracy for _multimodal fusion_ models on in-domain (ID) and cross-domain (CD) test sets. Results are reported for ResNeXt-50 and ViT-B/16 backbones under four fusion strategies: early channel stacking (St), concatenation of modality embeddings (C), cross-attention with a shared encoder (A1), and cross-attention with separate encoders (A2). ID–CD performance differences ($\Delta$) are also shown. The best and second-best scores in each column are indicated in **bold** and underlined, respectively.

| Modality / Fusion | mAP (ResNeXt) | | | mAP (ViT) | | | Accuracy (ResNeXt) | | | Accuracy (ViT) | | |
|---|---|---|---|---|---|---|---|---|---|---|---|---|
| | ID | CD | $\Delta$ | ID | CD | $\Delta$ | ID | CD | $\Delta$ | ID | CD | $\Delta$ |
| $EP_{ms}+S_{ms}+SDS_{ms}$ (St) | **0.571** | **0.495** | 0.076 | 0.534 | 0.463 | 0.070 | **0.875** | **0.867** | 0.008 | 0.834 | 0.823 | 0.011 |
| $EP_5+S_{1.5}+SDS_5$ (St) | 0.551 | 0.471 | 0.080 | **0.540** | 0.461 | 0.079 | 0.865 | 0.856 | 0.009 | 0.712 | 0.664 | 0.048 |
| $EP_{201}+S_{60}+SDS_{201}$ (St) | 0.552 | 0.480 | 0.072 | 0.532 | 0.468 | 0.064 | 0.858 | 0.852 | 0.006 | **0.851** | **0.840** | 0.011 |
| $EP_{ms}+S_{ms}+SDS_{ms}$ (C) | 0.505 | 0.451 | 0.053 | 0.508 | 0.450 | 0.058 | 0.822 | 0.836 | -0.015 | 0.817 | 0.806 | 0.011 |
| RGB+DEM (C) | 0.495 | 0.360 | 0.135 | 0.524 | 0.415 | 0.109 | 0.815 | 0.809 | 0.007 | 0.838 | 0.796 | 0.042 |
| RGB+DEM+$EP_{ms}+S_{ms}+SDS_{ms}$ (C) | 0.525 | 0.458 | 0.067 | 0.537 | 0.449 | 0.088 | 0.833 | 0.805 | 0.028 | 0.827 | 0.824 | 0.003 |
| $EP_{ms}+S_{ms}+SDS_{ms}$ (A1) | 0.474 | 0.442 | **0.033** | 0.488 | 0.456 | **0.032** | 0.747 | 0.758 | -0.011 | 0.750 | 0.752 | **-0.002** |
| RGB+DEM (A1) | 0.459 | 0.389 | 0.070 | 0.478 | 0.360 | 0.118 | 0.784 | 0.776 | 0.008 | 0.799 | 0.745 | 0.054 |
| $EP_{ms}+S_{ms}+SDS_{ms}$ (A2) | 0.474 | 0.442 | **0.033** | 0.392 | 0.333 | 0.059 | 0.747 | 0.758 | -0.011 | 0.452 | 0.402 | 0.050 |
| RGB+DEM (A2) | 0.464 | 0.389 | 0.075 | 0.486 | 0.388 | 0.098 | 0.795 | 0.793 | 0.002 | 0.795 | 0.775 | 0.020 |
| RGB+DEM+$EP_{ms}+S_{ms}+SDS_{ms}$ (A2) | 0.391 | 0.333 | 0.059 | 0.422 | 0.368 | 0.054 | 0.391 | 0.333 | 0.059 | 0.603 | 0.620 | -0.017 |

Table 17: Class-wise AUC scores for in-domain performance across single-modality, multi-scale fusion, and multimodal fusion models. Results are reported for ResNeXt-50 and ViT-B/16 backbones under four fusion strategies: early channel stacking (St), concatenation of modality embeddings (C), cross-attention with a shared encoder (A1), and cross-attention with separate encoders (A2). The best and second-best scores in each column are indicated in **bold** and underlined, respectively.

| Modality / Fusion | ResNeXt | | | | | | | ViT | | | | | | |
|---|---|---|---|---|---|---|---|---|---|---|---|---|---|---|
| | af1 | Qal | Qaf | Qat | Qc | Qca | Qr | af1 | Qal | Qaf | Qat | Qc | Qca | Qr |
| DEM | 0.845 | 0.832 | 0.820 | 0.887 | 0.964 | 0.922 | 0.910 | 0.663 | 0.771 | **0.926** | 0.871 | 0.956 | 0.923 | 0.888 |
| RGB | 0.834 | 0.713 | 0.684 | 0.815 | 0.912 | 0.857 | 0.886 | 0.816 | 0.679 | 0.744 | 0.780 | 0.891 | 0.834 | 0.805 |
| NIR | 0.816 | 0.698 | 0.782 | 0.793 | 0.907 | 0.866 | 0.842 | 0.760 | 0.664 | 0.797 | 0.799 | 0.886 | 0.816 | 0.763 |
| NHD | 0.549 | 0.655 | 0.682 | 0.782 | 0.618 | 0.630 | 0.697 | 0.497 | 0.571 | 0.441 | 0.354 | 0.506 | 0.502 | 0.505 |
| OSM | 0.807 | 0.586 | 0.702 | 0.586 | 0.708 | 0.627 | 0.557 | 0.505 | 0.484 | 0.693 | 0.606 | 0.5 | 0.513 | 0.487 |
| $EP_5$ | 0.837 | 0.805 | 0.845 | 0.845 | 0.947 | 0.905 | 0.920 | 0.791 | 0.783 | 0.838 | 0.865 | 0.914 | 0.885 | 0.903 |
| $EP_{11}$ | **0.868** | 0.816 | 0.833 | 0.888 | 0.936 | 0.905 | 0.902 | 0.778 | 0.781 | 0.834 | 0.882 | 0.891 | 0.889 | 0.898 |
| $EP_{21}$ | 0.856 | 0.807 | 0.842 | 0.883 | 0.945 | 0.908 | 0.900 | 0.783 | 0.776 | 0.799 | 0.858 | 0.888 | 0.885 | 0.880 |
| $EP_{51}$ | 0.860 | 0.825 | 0.827 | 0.870 | 0.921 | 0.906 | 0.924 | 0.794 | 0.766 | 0.791 | 0.858 | 0.877 | 0.888 | 0.870 |
| $EP_{101}$ | 0.853 | 0.806 | 0.759 | 0.886 | 0.904 | 0.904 | 0.890 | 0.757 | 0.751 | 0.758 | 0.860 | 0.850 | 0.884 | 0.874 |
| $EP_{201}$ | 0.846 | 0.812 | 0.844 | 0.879 | 0.894 | 0.894 | 0.904 | 0.734 | 0.750 | 0.756 | 0.830 | 0.789 | 0.872 | 0.864 |
| $PlC_{1.5}$ | 0.440 | 0.491 | 0.610 | 0.515 | 0.513 | 0.514 | 0.516 | 0.438 | 0.509 | 0.719 | 0.610 | 0.575 | 0.725 | 0.645 |
| $PlC_3$ | 0.501 | 0.501 | 0.500 | 0.500 | 0.501 | 0.501 | 0.500 | 0.445 | 0.494 | 0.769 | 0.675 | 0.499 | 0.773 | 0.689 |
| $PlC_6$ | 0.459 | 0.516 | 0.491 | 0.497 | 0.455 | 0.505 | 0.490 | 0.451 | 0.478 | 0.746 | 0.712 | 0.668 | 0.719 | 0.649 |
| $PlC_{15}$ | 0.526 | 0.505 | 0.362 | 0.387 | 0.547 | 0.476 | 0.500 | 0.466 | 0.523 | 0.655 | 0.620 | 0.578 | 0.575 | 0.505 |
| $PlC_{30}$ | 0.517 | 0.490 | 0.604 | 0.473 | 0.501 | 0.524 | 0.465 | 0.469 | 0.567 | 0.650 | 0.515 | 0.531 | 0.529 | 0.465 |
| $PlC_{60}$ | 0.462 | 0.413 | 0.617 | 0.414 | 0.479 | 0.494 | 0.439 | 0.461 | 0.627 | 0.620 | 0.382 | 0.524 | 0.482 | 0.402 |
| $PrC_{1.5}$ | 0.465 | 0.566 | 0.569 | 0.473 | 0.564 | 0.516 | 0.724 | 0.444 | 0.545 | 0.546 | 0.236 | 0.501 | 0.347 | 0.233 |
| $PrC_3$ | 0.549 | 0.555 | 0.324 | 0.537 | 0.341 | 0.554 | 0.539 | 0.545 | 0.501 | 0.630 | 0.400 | 0.420 | 0.613 | 0.508 |
| $PrC_6$ | 0.526 | 0.494 | 0.445 | 0.503 | 0.472 | 0.539 | 0.579 | 0.493 | 0.602 | 0.487 | 0.190 | 0.541 | 0.224 | 0.186 |
| $PrC_{15}$ | 0.443 | 0.423 | 0.602 | 0.522 | 0.145 | 0.377 | 0.567 | 0.501 | 0.429 | 0.477 | 0.378 | 0.501 | 0.499 | 0.476 |
| $PrC_{30}$ | 0.515 | 0.432 | 0.465 | 0.608 | 0.530 | 0.681 | 0.640 | 0.501 | 0.341 | 0.523 | 0.845 | 0.512 | 0.738 | 0.833 |
| $PrC_{60}$ | 0.482 | 0.499 | 0.494 | 0.244 | 0.473 | 0.474 | 0.253 | 0.511 | 0.326 | 0.558 | 0.859 | 0.601 | 0.682 | 0.846 |
| $S_{1.5}$ | 0.863 | 0.800 | 0.813 | 0.870 | 0.968 | 0.905 | 0.910 | 0.794 | 0.748 | 0.853 | 0.854 | 0.974 | 0.900 | 0.864 |
| $S_3$ | 0.816 | 0.805 | 0.840 | 0.870 | 0.971 | 0.915 | 0.908 | 0.770 | 0.759 | 0.772 | 0.829 | 0.975 | 0.910 | 0.868 |
| $S_6$ | 0.778 | 0.809 | 0.764 | 0.877 | 0.974 | 0.921 | 0.905 | 0.718 | 0.765 | 0.809 | 0.853 | 0.975 | 0.910 | 0.803 |
| $S_{15}$ | 0.648 | 0.788 | 0.842 | 0.873 | 0.966 | **0.926** | 0.842 | 0.641 | 0.750 | 0.826 | 0.796 | 0.974 | 0.908 | 0.789 |
| $S_{30}$ | 0.619 | 0.750 | 0.803 | 0.831 | 0.957 | 0.912 | 0.807 | 0.623 | 0.707 | 0.791 | 0.725 | 0.947 | 0.869 | 0.696 |
| $S_{60}$ | 0.416 | 0.535 | 0.681 | 0.595 | 0.838 | 0.815 | 0.324 | 0.626 | 0.666 | 0.818 | 0.750 | 0.909 | 0.880 | 0.738 |
| $SDS_5$ | 0.855 | 0.733 | 0.789 | 0.860 | 0.944 | 0.890 | 0.883 | 0.772 | 0.665 | 0.800 | 0.757 | 0.921 | 0.833 | 0.751 |
| $SDS_{11}$ | 0.839 | 0.751 | 0.774 | 0.866 | 0.946 | 0.877 | 0.871 | 0.792 | 0.671 | 0.817 | 0.757 | 0.933 | 0.853 | 0.800 |
| $SDS_{21}$ | 0.842 | 0.750 | 0.842 | 0.841 | 0.953 | 0.889 | 0.860 | 0.769 | 0.685 | 0.853 | 0.767 | 0.934 | 0.837 | 0.816 |
| $SDS_{51}$ | 0.832 | 0.719 | 0.851 | 0.800 | 0.951 | 0.883 | 0.852 | 0.675 | 0.620 | 0.777 | 0.684 | 0.889 | 0.759 | 0.689 |
| $SDS_{101}$ | 0.814 | 0.732 | 0.860 | 0.813 | 0.964 | 0.882 | 0.874 | 0.659 | 0.608 | 0.804 | 0.659 | 0.891 | 0.751 | 0.655 |
| $SDS_{201}$ | 0.802 | 0.679 | 0.812 | 0.833 | 0.967 | 0.897 | 0.870 | 0.633 | 0.605 | 0.855 | 0.666 | 0.913 | 0.741 | 0.729 |
| $EP_{ms}$ (St) | 0.823 | 0.824 | 0.734 | 0.878 | 0.945 | 0.911 | 0.917 | 0.823 | 0.824 | 0.734 | 0.878 | 0.945 | 0.911 | **0.917** |
| $PlC_{ms}$ (St) | 0.504 | 0.500 | 0.641 | 0.501 | 0.514 | 0.500 | 0.514 | 0.504 | 0.500 | 0.641 | 0.501 | 0.514 | 0.500 | 0.514 |
| $PrC_{ms}$ (St) | 0.494 | 0.653 | 0.567 | 0.721 | 0.628 | 0.791 | 0.201 | 0.494 | 0.653 | 0.567 | 0.721 | 0.628 | 0.791 | 0.201 |
| $S_{ms}$ (St) | 0.863 | 0.787 | 0.760 | 0.870 | 0.962 | 0.911 | 0.900 | **0.863** | 0.787 | 0.760 | 0.870 | 0.962 | 0.911 | 0.900 |
| $SDS_{ms}$ (St) | 0.839 | 0.766 | **0.917** | 0.876 | 0.964 | 0.898 | 0.889 | 0.839 | 0.766 | 0.917 | 0.876 | 0.964 | 0.898 | 0.889 |
| $EP_{ms}$ (A1) | 0.500 | 0.500 | 0.500 | 0.500 | 0.500 | 0.500 | 0.500 | 0.500 | 0.500 | 0.500 | 0.500 | 0.500 | 0.500 | 0.500 |
| $PlC_{ms}$ (A1) | 0.500 | 0.500 | 0.500 | 0.500 | 0.500 | 0.500 | 0.500 | 0.500 | 0.500 | 0.500 | 0.500 | 0.500 | 0.500 | 0.500 |
| $PrC_{ms}$ (A1) | 0.500 | 0.500 | 0.500 | 0.500 | 0.500 | 0.500 | 0.500 | 0.500 | 0.500 | 0.500 | 0.500 | 0.500 | 0.500 | 0.500 |
| $S_{ms}$ (A1) | 0.499 | 0.501 | 0.500 | 0.500 | 0.501 | 0.499 | 0.500 | 0.499 | 0.501 | 0.500 | 0.500 | 0.501 | 0.499 | 0.500 |
| $SDS_{ms}$ (A1) | 0.552 | 0.576 | 0.801 | 0.602 | 0.679 | 0.540 | 0.580 | 0.552 | 0.576 | 0.801 | 0.602 | 0.679 | 0.540 | 0.580 |
| $EP_{ms}+S_{ms}+SDS_{ms}$ (St) | 0.866 | **0.840** | 0.790 | 0.858 | **0.975** | 0.913 | **0.933** | 0.780 | 0.772 | 0.864 | 0.847 | **0.976** | 0.890 | 0.890 |
| $EP_5+S_{1.5}+SDS_5$ (St) | 0.845 | 0.797 | 0.712 | 0.829 | 0.964 | 0.904 | 0.886 | 0.837 | **0.803** | 0.858 | 0.884 | 0.974 | 0.912 | 0.901 |
| $EP_{201}+S_{60}+SDS_{201}$ (St) | 0.846 | 0.802 | 0.840 | **0.903** | 0.961 | 0.911 | 0.933 | 0.752 | 0.799 | 0.848 | 0.856 | 0.967 | **0.937** | 0.905 |
| $EP_{ms}+S_{ms}+SDS_{ms}$ (C) | 0.723 | 0.802 | 0.746 | 0.809 | 0.959 | 0.879 | 0.885 | 0.728 | 0.720 | 0.871 | 0.816 | 0.969 | 0.890 | 0.898 |
| RGB+DEM (C) | 0.821 | 0.708 | 0.804 | 0.803 | 0.871 | 0.845 | 0.803 | 0.800 | 0.756 | 0.874 | **0.899** | 0.949 | 0.901 | 0.911 |
| RGB+DEM+$EP_{ms}$+$S_{ms}$+$SDS_{ms}$ (C) | 0.837 | 0.774 | 0.842 | 0.827 | 0.963 | 0.899 | 0.860 | 0.746 | 0.755 | 0.875 | 0.878 | 0.975 | 0.910 | 0.921 |
| $EP_{ms}+S_{ms}+SDS_{ms}$ (A1) | 0.486 | 0.575 | 0.726 | 0.641 | 0.930 | 0.784 | 0.599 | 0.698 | 0.623 | 0.831 | 0.660 | 0.961 | 0.879 | 0.785 |
| RGB+DEM (A1) | 0.687 | 0.476 | 0.747 | 0.762 | 0.837 | 0.801 | 0.692 | 0.711 | 0.629 | 0.813 | 0.815 | 0.886 | 0.842 | 0.811 |
| $EP_{ms}+S_{ms}+SDS_{ms}$ (A2) | 0.486 | 0.575 | 0.726 | 0.641 | 0.930 | 0.784 | 0.599 | 0.500 | 0.500 | 0.623 | 0.534 | 0.500 | 0.500 | 0.494 |
| RGB+DEM (A2) | 0.752 | 0.617 | 0.780 | 0.816 | 0.825 | 0.786 | 0.764 | 0.704 | 0.692 | 0.856 | 0.806 | 0.930 | 0.841 | 0.839 |
| RGB+DEM+$EP_{ms}$+$S_{ms}$+$SDS_{ms}$ (A2) | 0.500 | 0.500 | 0.500 | 0.500 | 0.500 | 0.500 | 0.500 | 0.501 | 0.477 | 0.768 | 0.557 | 0.499 | 0.753 | 0.626 |

Table 18: Class-wise AUC for cross-domain performance across single-modality, multi-scale fusion, and multimodal fusion models. Results are reported for ResNeXt-50 and ViT-B/16 backbones under four fusion strategies: early channel stacking (St), concatenation of modality embeddings (C), cross-attention with a shared encoder (A1), and cross-attention with separate encoders (A2). Best and second-best scores in each column are indicated in **bold** and underlined, respectively.

| Modality / Fusion | ResNeXt | | | | | | | ViT | | | | | | |
|---|---|---|---|---|---|---|---|---|---|---|---|---|---|---|
| | af1 | Qal | Qaf | Qat | Qc | Qca | Qr | af1 | Qal | Qaf | Qat | Qc | Qca | Qr |
| DEM | 0.804 | 0.613 | 0.612 | 0.472 | 0.969 | 0.907 | 0.733 | 0.587 | 0.549 | 0.379 | 0.210 | 0.958 | 0.947 | 0.710 |
| RGB | 0.757 | 0.576 | 0.403 | 0.486 | 0.654 | 0.515 | 0.507 | 0.575 | 0.527 | 0.782 | 0.650 | 0.270 | 0.381 | 0.494 |
| NIR | 0.733 | 0.519 | 0.490 | 0.550 | 0.703 | 0.824 | 0.727 | 0.502 | 0.578 | 0.474 | 0.641 | 0.466 | 0.348 | 0.554 |
| NHD | 0.556 | 0.642 | 0.630 | 0.722 | 0.485 | 0.494 | 0.504 | 0.494 | 0.506 | 0.538 | 0.498 | 0.516 | 0.510 | 0.618 |
| OSM | 0.833 | 0.518 | 0.479 | 0.572 | 0.624 | 0.586 | 0.496 | 0.503 | 0.5 | 0.543 | 0.553 | 0.5 | 0.505 | 0.584 |
| $EP_5$ | 0.769 | 0.635 | 0.782 | **0.847** | 0.291 | 0.352 | 0.399 | 0.764 | 0.651 | 0.626 | 0.622 | 0.860 | 0.882 | 0.757 |
| $EP_{11}$ | 0.790 | 0.687 | 0.801 | 0.763 | 0.463 | 0.563 | 0.662 | 0.763 | 0.698 | 0.734 | 0.667 | 0.807 | 0.870 | 0.840 |
| $EP_{21}$ | 0.818 | 0.700 | 0.846 | 0.746 | 0.392 | 0.668 | 0.694 | 0.778 | 0.696 | 0.725 | 0.662 | 0.817 | 0.842 | 0.796 |
| $EP_{51}$ | 0.821 | 0.676 | 0.769 | 0.778 | 0.409 | 0.519 | 0.672 | 0.779 | 0.633 | 0.798 | 0.684 | 0.771 | 0.851 | 0.786 |
| $EP_{101}$ | 0.851 | 0.716 | 0.726 | 0.769 | 0.621 | 0.748 | 0.742 | 0.745 | 0.633 | 0.815 | 0.629 | 0.759 | 0.842 | 0.789 |
| $EP_{201}$ | 0.786 | 0.737 | 0.805 | 0.752 | 0.573 | 0.697 | 0.717 | 0.698 | 0.676 | 0.821 | 0.729 | 0.718 | 0.818 | 0.701 |
| $PlC_{1.5}$ | 0.492 | 0.501 | 0.599 | 0.548 | 0.487 | 0.509 | 0.453 | 0.514 | 0.340 | 0.650 | 0.518 | 0.561 | 0.792 | 0.752 |
| $PlC_3$ | 0.500 | 0.500 | 0.500 | 0.500 | 0.500 | 0.500 | 0.500 | 0.511 | 0.305 | 0.733 | 0.638 | 0.500 | 0.791 | 0.819 |
| $PlC_6$ | 0.517 | 0.480 | 0.529 | 0.478 | 0.474 | 0.492 | 0.426 | 0.530 | 0.304 | 0.758 | 0.701 | 0.627 | 0.703 | 0.766 |
| $PlC_{15}$ | 0.511 | 0.464 | 0.275 | 0.397 | 0.557 | 0.497 | 0.511 | 0.517 | 0.470 | 0.711 | 0.600 | 0.537 | 0.532 | 0.442 |
| $PlC_{30}$ | 0.513 | 0.514 | 0.324 | 0.516 | 0.497 | 0.472 | 0.454 | 0.517 | 0.527 | 0.809 | 0.512 | 0.536 | 0.527 | 0.349 |
| $PlC_{60}$ | 0.510 | 0.472 | 0.899 | 0.537 | 0.465 | 0.503 | 0.311 | 0.501 | 0.562 | 0.831 | 0.515 | 0.554 | 0.523 | 0.285 |
| $PrC_{1.5}$ | 0.426 | 0.559 | 0.263 | 0.418 | 0.679 | 0.710 | 0.559 | 0.412 | 0.633 | 0.219 | 0.362 | 0.500 | 0.592 | 0.404 |
| $PrC_3$ | 0.597 | 0.379 | 0.797 | 0.612 | 0.277 | 0.363 | 0.614 | 0.574 | 0.508 | 0.507 | 0.448 | 0.372 | 0.539 | 0.505 |
| $PrC_6$ | 0.498 | 0.490 | 0.408 | 0.414 | 0.478 | 0.491 | 0.459 | 0.417 | 0.644 | 0.348 | 0.468 | 0.584 | 0.393 | 0.256 |
| $PrC_{15}$ | 0.493 | 0.493 | 0.426 | 0.551 | 0.136 | 0.248 | 0.438 | 0.500 | 0.458 | 0.476 | 0.496 | 0.500 | 0.500 | 0.482 |
| $PrC_{30}$ | 0.506 | 0.448 | 0.150 | 0.505 | 0.552 | 0.631 | 0.646 | 0.532 | 0.428 | 0.566 | 0.528 | 0.463 | 0.664 | 0.867 |
| $PrC_{60}$ | 0.467 | 0.543 | 0.464 | 0.435 | 0.431 | 0.429 | 0.225 | 0.534 | 0.424 | 0.569 | 0.574 | 0.573 | 0.612 | **0.905** |
| $S_{1.5}$ | 0.863 | 0.737 | 0.611 | 0.754 | 0.975 | 0.915 | 0.801 | 0.759 | 0.579 | 0.646 | 0.667 | 0.981 | 0.923 | 0.778 |
| $S_3$ | 0.781 | 0.731 | 0.531 | 0.696 | 0.976 | 0.922 | 0.815 | 0.683 | 0.563 | 0.528 | 0.530 | 0.981 | 0.937 | 0.772 |
| $S_6$ | 0.713 | 0.704 | **0.889** | 0.706 | 0.976 | 0.924 | 0.717 | 0.621 | 0.569 | 0.646 | 0.708 | 0.981 | 0.941 | 0.509 |
| $S_{15}$ | 0.625 | 0.619 | 0.674 | 0.665 | 0.974 | 0.936 | 0.718 | 0.529 | 0.551 | **0.964** | 0.477 | 0.971 | 0.952 | 0.673 |
| $S_{30}$ | 0.550 | 0.549 | 0.746 | 0.537 | 0.965 | **0.945** | 0.675 | 0.533 | 0.559 | 0.704 | 0.372 | 0.945 | 0.959 | 0.859 |
| $S_{60}$ | 0.467 | 0.545 | 0.541 | 0.365 | 0.802 | 0.890 | 0.435 | 0.524 | 0.533 | 0.607 | 0.348 | 0.919 | **0.962** | 0.842 |
| $SDS_5$ | 0.858 | 0.637 | 0.805 | 0.737 | 0.963 | 0.886 | 0.744 | 0.776 | 0.561 | 0.503 | 0.593 | 0.958 | 0.864 | 0.739 |
| $SDS_{11}$ | **0.861** | 0.671 | 0.587 | 0.701 | 0.971 | 0.905 | 0.804 | 0.762 | 0.538 | 0.556 | 0.631 | 0.957 | 0.863 | 0.753 |
| $SDS_{21}$ | 0.838 | 0.673 | 0.749 | 0.794 | 0.969 | 0.869 | 0.613 | 0.741 | 0.543 | 0.658 | 0.694 | 0.952 | 0.853 | 0.704 |
| $SDS_{51}$ | 0.822 | 0.649 | 0.608 | 0.605 | 0.959 | 0.834 | 0.749 | 0.670 | 0.515 | 0.511 | 0.673 | 0.943 | 0.824 | 0.686 |
| $SDS_{101}$ | 0.809 | 0.611 | 0.443 | 0.788 | 0.960 | 0.886 | 0.795 | 0.656 | 0.474 | 0.491 | 0.644 | 0.954 | 0.871 | 0.677 |
| $SDS_{201}$ | 0.752 | 0.579 | 0.503 | 0.645 | 0.964 | 0.804 | 0.744 | 0.641 | 0.479 | 0.477 | 0.640 | 0.942 | 0.870 | 0.647 |
| $EP_{ms}$ (St) | 0.769 | 0.722 | 0.828 | 0.722 | 0.603 | 0.701 | 0.671 | 0.769 | **0.722** | 0.828 | 0.722 | 0.603 | 0.701 | 0.671 |
| $PlC_{ms}$ (St) | 0.479 | 0.524 | 0.603 | 0.489 | 0.553 | 0.567 | 0.432 | 0.479 | 0.524 | 0.603 | 0.489 | 0.553 | 0.567 | 0.432 |
| $PrC_{ms}$ (St) | 0.496 | 0.567 | 0.301 | 0.440 | 0.687 | 0.788 | 0.202 | 0.496 | 0.567 | 0.301 | 0.440 | 0.687 | 0.788 | 0.202 |
| $S_{ms}$ (St) | 0.881 | 0.711 | 0.643 | 0.741 | **0.977** | 0.915 | 0.759 | **0.881** | 0.711 | 0.643 | **0.741** | 0.977 | 0.915 | 0.759 |
| $SDS_{ms}$ (St) | 0.843 | 0.679 | 0.629 | 0.762 | 0.966 | 0.889 | 0.777 | 0.843 | 0.679 | 0.629 | 0.762 | 0.966 | 0.889 | 0.777 |
| $EP_{ms}$ (A1) | 0.500 | 0.500 | 0.500 | 0.500 | 0.500 | 0.500 | 0.500 | 0.500 | 0.500 | 0.500 | 0.500 | 0.500 | 0.500 | 0.500 |
| $PlC_{ms}$ (A1) | 0.500 | 0.500 | 0.500 | 0.500 | 0.500 | 0.500 | 0.500 | 0.500 | 0.500 | 0.500 | 0.500 | 0.500 | 0.500 | 0.500 |
| $PrC_{ms}$ (A1) | 0.500 | 0.500 | 0.500 | 0.500 | 0.500 | 0.500 | 0.500 | 0.500 | 0.500 | 0.500 | 0.500 | 0.500 | 0.500 | 0.500 |
| $S_{ms}$ (A1) | 0.500 | 0.500 | 0.500 | 0.500 | 0.500 | 0.500 | 0.500 | 0.500 | 0.500 | 0.500 | 0.500 | 0.500 | 0.500 | 0.500 |
| SDS (A1) | 0.558 | 0.592 | 0.699 | 0.679 | 0.626 | 0.602 | 0.568 | 0.558 | 0.592 | 0.699 | 0.679 | 0.626 | 0.602 | 0.568 |
| $EP_{ms}$+$S_{ms}$+$SDS_{ms}$ (St) | 0.857 | **0.760** | 0.612 | 0.736 | 0.972 | 0.914 | 0.792 | 0.734 | 0.586 | 0.740 | 0.650 | **0.982** | 0.922 | 0.805 |
| $EP_5$+$S_{1.5}$+$SDS_5$ (St) | 0.860 | 0.638 | 0.735 | 0.760 | 0.960 | 0.899 | 0.833 | 0.848 | 0.683 | 0.685 | 0.697 | 0.980 | 0.922 | 0.803 |
| $EP_{201}$+$S_{60}$+$SDS_{201}$ (St) | 0.859 | 0.717 | 0.699 | 0.685 | 0.962 | 0.911 | **0.855** | 0.657 | 0.587 | 0.748 | 0.646 | 0.976 | **0.962** | 0.879 |
| $EP_{ms}$+$S_{ms}$+$SDS_{ms}$ (C) | 0.701 | 0.693 | 0.498 | 0.689 | 0.962 | 0.902 | 0.804 | 0.679 | 0.577 | 0.633 | 0.582 | 0.973 | 0.938 | 0.765 |
| RGB+DEM (C) | 0.788 | 0.460 | 0.173 | 0.406 | 0.661 | 0.621 | 0.635 | 0.752 | 0.554 | 0.545 | 0.611 | 0.930 | 0.923 | 0.732 |
| RGB+DEM+$EP_{ms}$+$S_{ms}$+$SDS_{ms}$ (C) | 0.841 | 0.644 | 0.452 | 0.493 | 0.964 | 0.946 | 0.833 | 0.660 | 0.540 | 0.687 | 0.594 | 0.965 | 0.933 | 0.825 |
| $EP_{ms}$+$S_{ms}$+$SDS_{ms}$ (A1) | 0.555 | 0.525 | 0.674 | 0.552 | 0.921 | 0.907 | 0.816 | 0.653 | 0.483 | 0.500 | 0.377 | 0.973 | 0.955 | 0.805 |
| RGB+DEM (A1) | 0.708 | 0.527 | 0.274 | 0.130 | 0.836 | 0.740 | 0.647 | 0.671 | 0.513 | 0.271 | 0.548 | 0.916 | 0.901 | 0.531 |
| $EP_{ms}$+$S_{ms}$+$SDS_{ms}$ (A2) | 0.555 | 0.525 | 0.674 | 0.552 | 0.921 | 0.907 | 0.816 | 0.500 | 0.500 | 0.362 | 0.497 | 0.500 | 0.500 | 0.505 |
| RGB+DEM (A2) | 0.743 | 0.482 | 0.325 | 0.499 | 0.905 | 0.835 | 0.695 | 0.688 | 0.498 | 0.695 | 0.676 | 0.941 | 0.894 | 0.677 |
| RGB+DEM+$EP_{ms}$+$S_{ms}$+$SDS_{ms}$ (A2) | 0.500 | 0.500 | 0.500 | 0.500 | 0.500 | 0.500 | 0.500 | 0.515 | 0.451 | 0.250 | 0.350 | 0.500 | 0.860 | 0.670 |

