# OpenReview forum: "EarthScape: A Multimodal Dataset for Surficial Geologic Mapping and Earth Surface Analysis"
_ICLR.cc/2026/Conference — Submitted to ICLR 2026_

### Official Review · Reviewer_QJiJ · 2025-10-23

**Soundness:** 3
**Presentation:** 2
**Contribution:** 2
**Rating:** 2
**Confidence:** 3

**Summary:**

The paper introduces EarthScape, a multimodal, multi-scale benchmark dataset designed for surface geology (SG) mapping and broader surface analysis tasks. It integrates diverse data sources, including RGB and NIR imagery, digital elevation models (DEMs), DEM-derived shape features at multiple scales, and vector GIS layers for transportation and hydrological networks. The authors also provide baselines for unimodal, multi-scale, and multimodal configurations using a custom model architecture as well as some FMs using the multi-label classification task.

**Strengths:**

-	Solid motivation and well-grounded related work.
-	Inclusion of a geographic hold-out set to evaluate generalization.
-	Novel and interesting dataset; no similar resource appears to exist according to the related work.
-	Extensive experimental evaluation, with many results (primarily reported in the appendix).

**Weaknesses:**

-	The main paper evaluates only one custom architecture. While additional tests with EO foundation models appear in the appendix, this critical section lacks detailed information and ablation studies.
-	All experiments are confined to Kentucky, leaving uncertainty about performance in other geographic regions.
-	The paper asserts applicability to segmentation, multi-label classification, and regression, but only provides baselines for the multi-label task.
-	The main paper includes only a single results table and omits comprehensive ablation studies. Visual diagrams would strengthen the insights presented in Section 4.2.
-	Model names in tables are inadequately explained, making them difficult to interpret.

**Questions:**

-	Several prior works on ML for SG mapping are cited, but the Related Work section does not mention any specific SG datasets. What datasets were used in those ML papers? Clarifying this would help position EarthScape relative to existing resources.
-	The dataset merges multiple modalities, including several DEM-derived features that may carry overlapping information. Which modalities contribute most to performance, and which appear redundant? Figure 10 suggests some redundancy—why not filter out modalities that do not improve results? An ablation or modality importance analysis would strengthen the paper.

---

> ### Author Response · Authors · 2025-11-26
>
> **Existing SG datasets:** To our knowledge, there is no publicly available benchmark for SG mapping, which is precisely what motivates EarthScape (ES). Prior ML studies we cite rely on locally curated inventories that are not released as standardized datasets and typically cover small areas with limited modalities. Even if accessible, these resources focus on discrete events or single regions, rather than continuous SG mapping with co‑registered DEM, multi‑scale terrain derivatives, imagery, and vector layers. ES is designed to fill this gap. In the revised paper, we will better emphasize this to position ES’s contribution relative to prior work.
>
> **Modality importance:** We agree that understanding which modalities matter is important. Single‑modality and multi‑scale experiments (4.2, A.6) show that shape‑centric terrain derivatives consistently outperform RGB, NIR, and raw DEM in both in‑ and cross‑domain tests (Table 5, Table 8). Multi-scale S and SDS, in particular, achieve the smallest macro-F1 domain shift of 0.637 / 0.594 and 0.636 / 0.588, respectively (Table 8). Tables 14 and 15 show that there is no single scale that outperforms across individual classes. PlC, PrC, NHD, and OSM contribute less to performance for our baseline, but we retain them because (i) they are informative for some classes and backbones and (ii) they are realistic components of operational workflows and may be leveraged by future models and tasks. We will highlight EP+S+SDS as our strongest subset in the revised paper, while also explaining our rationale to keep the full modality set.
>
> **SGMap‑Net vs. SOTA comparisons:** SGMap‑Net was intentionally designed as a simple, transparent baseline so that the effects of modality, scale, and fusion strategy can be better isolated and interpreted. Within SGMap‑Net, we already perform ablations over single‑modality, multi‑scale, and multimodal configurations using early channel stacking, mid-level concatenation, and attention‑based fusion (4.2, A.6). To contextualize this baseline, we compared SatMAE, SatMAE++, DOFA, and Panopticon on ES (4.2, A.6.5). For example, SatMAE++ attains macro‑F1 0.656 in‑domain but drops to 0.454 cross‑domain, whereas SGMap‑Net with multi‑scale EP+S+SDS (stacking) achieves 0.657 / 0.598 with a much smaller gap (Table 16). We agree that positioning these models more clearly in the main text would strengthen the contribution, and we will expand their discussion and results in the revised main paper.
>
> **Geographic generalization:** ES v1.0 currently covers two regions in the central US, which are both governed by similar geologic processes of fluvial deposition, hillslope transport, and in‑situ weathering (3, A.2.2, A.2.3, Fig. 3, Table 2). Our cross‑domain experiments already show a significant domain shift between both areas, but multimodal, multi‑scale fusion of EP, S, and SDS reduces the macro‑F1 drop from 0.271 to 0.059 (Table 1, Table 11). Our experiments and interpretations are founded on domain knowledge, and scoped to geologically similar settings rather than geographic proximity (A.2.2, A.2.3, Table 3). The roadmap in A.2.1 is a first step toward broader coverage of additional geologic environments. We will make this scope and roadmap more explicit in the revised paper.
>
> **Additional tasks:** ES includes full pixel‑wise SG masks and per‑patch class‑area statistics for all patches (3.2, 3.3), enabling segmentation and regression in addition to multi‑label classification. We focused on patch‑level multi‑label classification because it is the most direct formulation of SG mapping at the patch scale, aligns with our goal of analyzing modality and scale effects, and fits within the space constraints of this D&B submission. In the revised paper we will highlight that the same inputs and labels can be used directly for segmentation and regression baselines.
>
> **Main‑paper presentation:** Thank you for the suggestions. The full ablations already appear in A.6 (Tables 5-16, Figs. 10-11). In the revised paper, we will strengthen 4.2 by making the main modality and fusion trends more explicit in the text and by refining table captions to better explain our naming conventions (e.g., EP-ms = multi-scale elevation percentile; S-3 = slope at 3.05 m GSD; St/C/A1/A2 = stacking, concatenation, and attention variants). If space allows, we will also include a very compact summary figure.

---

### Official Review · Reviewer_YDrN · 2025-10-26

**Soundness:** 3
**Presentation:** 3
**Contribution:** 3
**Rating:** 4
**Confidence:** 3

**Summary:**

This paper introduces EarthScape, the first AI-ready multimodal dataset for surficial geologic mapping, comprising 31k 256×256 patches with 38 co-registered channels of RGB/NIR imagery, LiDAR DEM, multi-scale terrain derivatives, and hydro-infrastructure vectors. Using the lightweight SGMap-Net baseline, we systematically evaluate unimodal, multi-scale, and multimodal fusion strategies, demonstrating that slope-based features generalize best across geologically distinct regions. Multi-scale channel stacking of elevation percentile, slope, and slope-standard-deviation achieves the top macro-F1 (0.657) with the smallest domain shift (ΔF1 0.059), establishing a reproducible and extensible benchmark for automated surficial geologic mapping.

**Strengths:**

EarthScape pioneers the first AI-ready surficial-geologic dataset with 31 k rigorously aligned 38-channel image-terrain-vector patches, and its exhaustive single-/multi-scale and multimodal ablations fill the large-scale label gap while quantitatively demonstrating that slope-based features offer the strongest cross-domain generalization.

**Weaknesses:**

1. The coverage of the dataset is limited.
The dataset only includes data from two counties Warren and Hardin, missing other geomorphological zones such as glacial, arid, and tropical regions, which limits the external validity of cross-domain conclusions

2. The dataset has issues of imbalanced data categories.
Seven types of SG units were sampled according to their natural occurrence frequency. Minority classes (Qaf, Qat) have very few samples, which easily leads to model bias toward majority classes, affecting the precision and recall of minority classes.

3. The architecture of SGMap-Net is overly simple, without any novelty. The choice of aggregation method is not verified by experiments.

**Questions:**

See weakness.

---

> ### Author Response · Authors · 2025-11-26
>
> **Geographic coverage & external validity:** As you point out, EarthScape (ES) v1.0 does not yet cover all geological or environmental settings. The current focus on two regions ensures high-quality SG maps, consistent mapping standards, and tightly co‑registered modalities. This controlled setting allows us to isolate modality and scale effects, while still introducing a meaningful geographic shift. Our experiments show that terrain derivatives describing surface shape are more transferable across this shift than raw imagery or DEM inputs (4.2, A.6, Tables 5-15). We do not claim global generalization, but, rather, we view ES v1.0 as representative of geologically similar, non‑glaciated fluvial-colluvial-residuum settings within the Interior Low Plateaus and adjacent provinces (A.2.2, Fig. 3), with plausible analogs in regions such as the Ozark Plateau, Dinaric Alps, and parts of China.
>
> We agree that incorporating glacial, arid, and tropical environments will further broaden the benchmark’s relevance. At present, this is limited by the availability of high‑quality, modern 1:24k SG maps with compatible vector and LiDAR standards, rather than by any fundamental constraint of ES curation. By establishing this foundation and releasing an open, reproducible processing pipeline, ES is intended to support external contributions so that additional terrain settings can be integrated with consistent QC. The roadmap in A.2.1 already includes extending coverage into glaciated terrains, and we will emphasize this scope and expansion plan more clearly in the revised paper.
>
> **Class imbalance:** The lower precision and recall observed for minority classes (e.g., Qaf, Qat) reflect their genuine rarity and small spatial footprint in SG maps, not an artifact of sampling. Fig. 2B-D and Fig. 8 show that Qr dominates the pixel distribution, while Qaf/Qat occur infrequently and often occupy small proportions of each patch, leading to both inter‑class imbalance and intra‑patch sparsity. ES intentionally preserves this true distribution because realistic SG mapping and Earth surface analysis require models to operate under long‑tailed, multilabel conditions. In all experiments we use a fixed focal loss configuration (A.5) for consistent comparability; we also tested oversampling, which yielded unstable training. In the revised paper we will make this discussion more explicit and add concise quantitative imbalance metrics so users can design and evaluate their own rebalancing strategies.
>
> **SGMap-Net & aggregation:** SGMap‑Net was intentionally designed to be simple, transparent, and reproducible, consistent with the expectations of a D&B contribution, rather than to introduce a novel architecture. It incorporates modality‑aware fusion strategies chosen for ES’s inputs, and we empirically compare early channel stacking, mid‑level concatenation, and attention‑based fusion in both in‑domain and cross‑domain settings (4.2, A.6, Tables 8-13). For example, with EP+S+SDS, early stacking achieves macro‑F1 0.657 (in‑domain) / 0.598 (cross‑domain), mid‑level concatenation 0.596 / 0.569, and attention‑based fusion 0.561 / 0.532 (Table 11). Thus, stacking yields the highest in-domain performance, while mid‑level strategies slightly reduce the domain gap at the cost of peak performance.
>
> Comparisons with existing remote sensing foundation models (4.2, A.6.5, Table 16) further support this design. SatMAE++ attains 0.656 macro‑F1 in‑domain, but drops to 0.454 cross‑domain, whereas SGMap‑Net with multi‑scale terrain derivatives achieves 0.657 / 0.598 with a substantially smaller gap. This suggests that a simple, task‑tailored fusion of shape‑centric terrain features provides a strong and interpretable baseline on ES. In the revised paper we will expand the summary of SOTA comparisons in the main paper, and that SGMap‑Net is intended as a reference architecture, although ES is designed to support more sophisticated models in future work.

---

### Official Review · Reviewer_ASL5 · 2025-10-28

**Soundness:** 2
**Presentation:** 2
**Contribution:** 2
**Rating:** 4
**Confidence:** 3

**Summary:**

This paper introduces EarthScape, a multimodal dataset designed for surficial geologic (SG) mapping and broader Earth surface analysis. The dataset integrates digital elevation models (DEMs), RGB+NIR aerial imagery, multi-scale terrain derivatives (e.g., slope, curvature), and vector-based features for hydrology (NHD) and infrastructure (OSM). It is sourced from publicly available U.S. Geological Survey (USGS) SG maps covering limited regions of the U.S., processed into 256x256 patches with 38 co-registered channels. The authors provide a reproducible processing pipeline and establish baselines using standard segmentation models (e.g., U-Net variants) across unimodal, multi-scale, and multimodal setups. Key challenges highlighted include class imbalance, geographic heterogeneity, and the need for multimodal fusion. Experiments demonstrate that terrain derivatives are highly predictive, but cross-region generalization remains poor, positioning EarthScape as a benchmark for domain adaptation and fusion techniques.

**Strengths:**

1. The dataset addresses a practical gap in geospatial domain by focusing on surficial geology, which has applications in hazard mitigation, urban planning, and climate studies.

2. Integration of multi-scale terrain features and vector data adds a layer of physical interpretability, potentially benefiting downstream tasks beyond geology, such as autonomous navigation or medical imaging analogies.

3. The emphasis on reproducibility via an end-to-end pipeline and use of public sources is commendable, and the baselines provide a clear starting point for future comparisons.

**Weaknesses:**

1. Limited Novelty and Broader Contribution: Dataset papers are increasingly common in machine learning, particularly in remote sensing and geospatial domains. EarthScape, while tailored to SG mapping, does not introduce particularly unique elements—its multimodal fusion of imagery, DEMs, and derivatives echoes existing RS benchmarks, and the focus on surface morphology is not sufficiently differentiated from prior works on landslides or land cover. The "living resource" claim lacks specifics on expansion plans, and the dataset's U.S.-centric nature limits its global representativeness, potentially reducing its appeal as a general benchmark for multimodal learning or domain adaptation.

2. Scope and Scale Concerns: The dataset appears constrained in geographic diversity, relying on site-specific SG maps that may not capture global variability in geologic processes. Class imbalance and long-tail issues are mentioned but not quantified in detail, which is a missed opportunity to highlight uniqueness. Additionally, the processing pipeline, while reproducible, relies on standard tools, and the choice of 256x256 patches at fixed scales may not adequately address multi-resolution challenges in real-world RS tasks.

3. Baseline Experiments and Evaluation: The baselines seem preliminary, focusing on standard architectures without exploring state-of-the-art multimodal models (e.g., CLIP variants, Perceiver IO, or recent geospatial transformers like GeoCLIP). Results emphasize terrain predictiveness but lack ablation on modality contributions or comparisons to non-ML methods (e.g., traditional GIS-based mapping). Cross-region generalization is identified as a challenge, but without metrics like domain shift quantification (e.g., MMD or adversarial scores), it's unclear how EarthScape advances beyond existing datasets. Moreover, no discussion of computational efficiency or scalability for large-scale deployment.

4. Minor Issues: The writing occasionally overstates impact (e.g., "unusually challenging benchmark") without empirical evidence comparing difficulty to other datasets.

**Questions:**

1. Why were the baselines limited to basic configurations? Could you include comparisons to recent multimodal foundation models (e.g., Prithvi or SatMAE) or non-deep learning approaches to better contextualize performance?

2. How was class imbalance addressed in training (e.g., weighted losses, oversampling)? Did you evaluate sensitivity to patch size or scale selection in the terrain derivatives?

---

> ### Author Response · Authors · 2025-11-26
>
> **Novelty:** The goal for EarthScape (ES) is not geographic scale for its own sake, but to open up a task not represented in existing benchmarks. ES combines (i) 31,018 multimodal patches with 38 channels each, (ii) mulitlabel classes reflecting natural SG processes documented worldwide, and (iii) an explicit cross‑region split. To our knowledge, no prior benchmark integrates DEMs, multi‑scale terrain derivatives, imagery, and vector hydrology/infrastructure into a single dataset with geographic generalization. ES terrain features consistently outperform imagery-centric SOTA RS models, demonstrating a challenging and unsolved learning problem. The design is broad, supporting research in physically-grounded representation learning like environmental modeling, hazard assessment, autonomous navigation, or other domains benefiting from surface awareness.
>
> **Scope:** ES v1.0 covers two US-based regions, but the geologic processes and surface expressions in these regions are widely occurring and not limited to geography (A.2.2; analogs to Ozark Plateau, Dynaric Alps, China, etc.). We agree that additional diversity would further strengthen the benchmark, and our roadmap includes that expansion (A.2.1): v2.0 will nearly triple the patch count, and v3.0 will extend into glacial terrains. ES expansion requires careful preprocessing for integration, but is underway. We will emphasize the natural generalization of geologic processes and our expansion roadmap more explicitly in the revised paper.
>
> **“Living” resource:** ES follows a semantic versioning scheme with frozen releases, a CHANGELOG, modular expansion of new areas to preserve benchmarks, and community contributions. Details are already described in A.2.1, but we will clarify and expand these plans in the revised paper.
>
> **SOTA comparisons:** SGMap‑Net was intentionally designed as a lightweight, transparent baseline rather than a novel architecture, consistent with D&B expectations. We already present comparisons with recent foundation models, including SatMAE, SatMAE++, DOFA, and Panopticon (4.2, A.6.5, Table 16). SatMAE and SatMAE++ achieve strong in‑domain macro‑F1 (0.614, 0.656) but drop to 0.427 and 0.454 under cross‑region testing, whereas our best SGMap‑Net configuration reaches 0.657 in‑domain and 0.598 cross‑domain; DOFA and Panopticon perform worse on both sets. CLIP‑style models are less directly applicable here because SG units lack natural language supervision and ES channels are not natively supported by image-text models; however, we see this style of adaptation as a key use of this benchmark. Classical GIS approaches rely on hand‑crafted, site‑specific features, and their reproducibility as benchmarks is limited. We will make these points clearer and add SOTA results to the main paper to make them more visible.
>
> **Class imbalance:** ES’s long‑tailed structure is visible in Fig. 2B-D and Fig. 8, which show that class Qr dominates, while classes Qaf and Qat are rare and occupy small fractions of a patch. We address this in our experiments using a single focal loss configuration for comparability (4.1, A.5); oversampling was tested, but degraded stability. We appreciate the suggestion and will add additional visualizations and metrics that directly highlight ES's imbalance.
>
> **Patch size:** The 256x256 patch size (~390x390 m) was chosen to match identifying landforms and patterns common to 1:24K-scale mapping (patch grid includes valleys, hillslopes, floodplains, etc. visible in Fig. 4). The overlapping patch design allows users to construct larger context windows if desired. We will clarify this rationale and note that, although we fixed patch size for consistency, the released data and code support alternative patch sizes.
>
> **Terrain scale:** Multi‑resolution structure is handled through the terrain derivatives. S, PlC, PrC, EP, and SDS are computed at six ~logarithmic scales and upsampled to a common 1.52 m GSD grid (3.2, A.2.4), so each patch encodes both local and regional morphology. Our experiments (4.2, A.6) already provide an evaluation of scale sensitivity. No single scale performs best for all classes, and multi-scale fusion improves generalization. We will reinforce this in the revised paper.
>
> **Cross-generalization:** Domain shift is already quantified via in- vs. cross-domain performance. However, we will additionally provide simple MMD values of raw features to support interpretation.
>
> **Compute:** The revised paper will report parameter counts and FLOPs to provide a measure of computational cost.
>
> **Writing:** We will revise phrasing that overstates impact or difficulty.

---

### Official Review · Reviewer_ztAy · 2025-11-02

**Soundness:** 2
**Presentation:** 3
**Contribution:** 1
**Rating:** 0
**Confidence:** 5

**Summary:**

This paper proposes a dataset called EarthScape for surficial geological mapping using multimodal remote sensing data. There are many datasets for remote sensing and land cover mapping, disaster detection, and other anthropomorphic or single-event phenomena, there are few datasets like EarthScape for surface geology. The dataset covers sites in two counties in Kentucky. It includes aerial RGB+NIR imagery and LiDAR-derived DEMs with classification and segmentation labels, both provided by Kentucky agencies. The paper also proposes a model, SGMap-Net, to model the dataset. The paper shows experiments using the dataset for multiple fusion strategies and architecture choices and shows that SGMap-Net performs best overall.

**Strengths:**

- The paper is well written and easy to read.
- There is a gap in datasets for surface geology in the remote sensing + ML research space, which this dataset helps fill.
- There are not many multimodal becnhmark remote sensing datasets that include DEMs, especially where DEMs are an important variable, so this dataset helps fill that gap.

**Weaknesses:**

- The dataset is called EarthScape, implying a global scale, but only represents two counties in Kentucky. The paper says that more sites will be added by 2026, but it’s unclear how that will be done and this work does not seem complete or high enough impact in its current form (before the broader geographic scope).
    - The paper says the limitation of two regions in Kentucky reflects the availability of 1:24k scale SG maps, but it seems there are a ton available from the USGS: https://ngmdb.usgs.gov/mapview/?center=-112.161,38.452&zoom=9
    - If this is meant to be a “living resource” how will it be updated, versioned, and kept consistent for use as an effective research benchmark? It doesn’t seem like the planned pace of its expansion will keep up with multimodal research in ML.
- The paper states that one of the challenges of machine learning for surficial geologic maps is that the classes on the surface are continuous and geologic maps are often subjective depending on the expertise or preferences of the mapper. How is this class ambiguity handled in the dataset? It seems like this aspect is not addressed.
- The aerial and lidar datasets both come from Kentucky agencies, so it’s not clear how it will be expanded or applied to other regions where those same datasets may not exist.
- The paper states that the dataset is a challenging benchmark for multimodal surface-aware learning and will contribute to multimodal learning research with remote sensing, but the experiments don’t engage with any recent work in multimodal learning for remote sensing (for example, geospatial foundation models that have RGB+NIR and DEM input channels).

**Questions:**

- If all of the code for creating/reproducing the dataset is done, why will it take until end of 2026 to add more regions?
- What prevents the use of the many 1:24k scale SG maps on the USGS site?
- If this is meant to be a “living resource” how will it be updated, versioned, and kept consistent for use as an effective research benchmark?
- How is this class ambiguity stemming from subjective geologic mapping and continuous landforms/materials handled in the dataset?

---

> ### Author Response · Authors · 2025-11-26
>
> **Scope & expansion:** EarthScape (ES) v1.0 already contains 31,018 multimodal patches (38 channels) spanning two distinct areas, with an explicit cross-region split for geographic generalization. Even in its current form, ES offers is the only surface aware benchmark that we are aware of. While the preprocessing code is complete, adding new regions is not trivial: reconciling SG map schemas with ES’s process-based classes, assessment and repair of target raster topologies from converted GIS vector products, validating geologic alignment with LiDAR-quality DEMs and imagery, and documenting any residual uncertainty. This work cannot yet be reliably automated without compromising dataset integrity. Given the required domain-expert involvement, we therefore follow the roadmap described in A.2.1: v2.0 will nearly triple the patch count, and v3.0 will extend coverage beyond Kentucky. We will make this schedule and the significant role of expert QA/QC more explicit in the revised paper.
>
> **USGS maps:** Although the USGS database contains many 1:24K products, ES imposes stricter inclusion criteria: (i) dense SG coverage in the target region, (ii) standardized GIS vector maps, and (iii) produced with high-resolution LiDAR basemaps. In Kentucky alone, most entries tagged as “surficial” are actually bedrock-focused maps with limited SG information, created on pre-LiDAR topography, and lack a standardized and topologically-verified vector product. In practice, only a small subset of maps simultaneously satisfy (i)–(iii) and are immediately ingestible; others would require significant re-registration and re-interpretation. For these reasons, only a small subset of USGS maps are immediately usable, which we will tackle incrementally.
>
> **Class ambiguity:** Interpretation and variability is inherent in manual expert SG mapping, reflecting a genuine characteristic of the domain, rather than a weakness of the dataset. SG map subjectivity most often reflects differences in local stratigraphic nomenclature, rather than the underlying geologic material. ES intentionally mitigates this by adopting process-based classes to avoid these inconsistencies and to provide labels that are transferable across regions. Boundaries between classes may also be subjective in areas where identifying criteria are not continuous in space and must be extrapolated. We do not currently encode per-pixel uncertainty and is treated as realistic label noise that models must handle. We will clarify this labeling scheme and the role of boundary uncertainty in the revised paper.
>
> **“Living” resource:** EarthScape is versioned as a sequence of frozen releases (v1.0, v1.1, etc.) with semantic version numbers, a CHANGELOG, and separate modules for new regions (App. A.2.1). This ensures that each snapshot remains a stable benchmark, and users can choose either a fixed core benchmark (v1.0) or expanded variant.
>
> **Broader applicability:** Although ES v1.0 uses aerial RGB+NIR and LiDAR DEMs from Kentucky, the modalities themselves are widely available (e.g., national/global DEM programs, multispectral aerial or satellite imagery, and open hydrography/infrastructure vectors). Our preprocessing pipeline operates on co‑registered DEM, imagery, and vector products and can ingest other sensors by re-projection and resampling to the target grid, subject to sufficient resolution for 1:24k‑scale mapping. We will emphasize that the design is not limited to current geographic extent, and is intended to support all regions with compatible RS products.
>
> **SOTA comparisons:** We did evaluate several SOTA RS models, including SatMAE, SatMAE++, DOFA, and Panopticon (4.2, A.6.5, Table 16). For example, SatMAE and SatMAE++ achieve strong in-domain macro-F1 (0.614, 0.656) but drop sharply under cross-domain testing (0.427, 0.454), whereas the best SGMap-Net configuration attains 0.657 in-domain and 0.598 cross-domain. These results support our main conclusion that terrain-derived, multi-scale shape features are essential for SG mapping and that ES exposes failure modes of imagery-centric foundation models. We will move these comparisons from the supplemental into the main text so they are harder to miss.
>
> **“EarthScape” name:** Our intention with “EarthScape” is to emphasize Earth surface processes and multimodal surface morphology, not immediate global coverage. This is consistent with naming patterns such as Cityscapes and SpaceNet, which highlight subject matter rather than current geographic extent. We will clarify this in the revised paper.

---

### Author Response · Authors · 2025-12-03
**Revised Manuscript**

We appreciate the reviewers’ suggestions and feedback. We have revised the manuscript accordingly, and the key updates are summarized below. Overall, these revisions substantially strengthen clarity, rigor, and transparency, and directly address all major concerns while preserving EarthScape’s core contribution. We welcome the opportunity to make further revisions based on additional feedback.

**Geographic scope, expansion, and dataset stability:** We clarified why EarthScape v1.0 focuses on two regions: adding new areas requires domain-expert QA/QC to reconcile SG map schemas, validate raster topologies, and ensure alignment with DEMs and imagery. The revised paper now includes a clear, realistic roadmap for expansion, along with semantic versioning, frozen splits, and modular additions to ensure benchmark stability.

**Broader applicability and geologic representativeness:** We expanded discussion of how the v1.0 classes and surface processes generalize to similar geologic settings globally. Although the initial release targets non-glaciated fluvial–colluvial–residuum environments, the pipeline is not geographically constrained. The revision clarifies that the preprocessing workflow is sensor-agnostic and supports any coregistered SG map, DEM, imagery, and vector data via reprojection and resampling. The roadmap notes that additional geologic environments will be incorporated as high-quality data becomes available.

**SOTA comparisons and SGMap-Net clarification:** We moved all SOTA comparisons into the main paper and clarified that SGMap-Net is intentionally lightweight and transparent, designed to isolate modality, scale, and fusion effects without architectural novelty claims. We expanded discussion of fusion and backbone design choices to better explain how they interact with EarthScape’s multimodal structure.
Class imbalance, rarity, and labeling ambiguity: We improved discussion of EarthScape’s long-tailed SG distribution by adding clearer imbalance metrics, updated figures, and more explicit explanations of why the dataset preserves the true class frequencies and intra-patch sparsity. We clarified handling of class-boundary uncertainty and the rationale behind process-based classes, while retaining realistic label noise inherent to expert mapping.

**Modality importance, multi-scale structure, and full modality set:** We consolidated and clarified the rationale for including a comprehensive set of modalities and scales: different features contribute to different SG units, reflect operational workflows, and support future research on modality selection. The revised writing makes explicit why certain derivatives and scales remain part of the dataset even when their contributions vary across tasks.

**Patch size, scale, and domain-shift rationale:** We added a concise justification for the 256x256 patch size and expanded explanation of how multi-scale terrain derivatives encode both local and regional morphology. The revised paper also includes a simple MMD analysis to quantify domain shift and situate experimental results.

**Positioning relative to prior SG datasets and supported tasks:** We strengthened the discussion of prior work, clarifying that no publicly available, standardized SG mapping benchmark currently exists. We also highlight that EarthScape supports segmentation and regression tasks in addition to multilabel classification.

**Presentation and clarity:** The main paper now contains clearer figures, tables, captions, and terminology, along with a more intuitive organization of the dataset, experimental results, and appendices. We also toned down any phrasing that overstated impact or difficulty.

---

### Meta-Review · Area_Chair_q82a · 2026-01-10

**Summary:**

Four expert reviewers agree on rejection with two strong votes for rejection rating 0 (ztAy) and 2 (QJiJ). While reviewers acknowledge the gap in machine learning for remote sensing concerning geological surface modeling, which the submission is meant to address, they agree on important limitations that prevent acceptance at this time. These issues include (1) geographic scope limited to Kentucky, (2) missing comparisons to current existing models including multi-modal remote sensing models as well as non-ML GIS methods, (3) imprecision about potentially subjective annotations about mapping, and (4) unsubstantiated potential for extension and maintenance as a "living resource". On the plus side, there is nevertheless appreciation for more work on geological surface modeling / use of DEMs in machine learning-based approaches, and for the downstream applications of hazard mitigation and climate studies, so the authors are encouraged to consider the feedback on scope and characterization of the subjectivity of annotations and submit to future venues.

**Reviewer Concerns:**

- Restricted geographic scope (ztAy, ASL5, YDrN, QJiJ): The proposed benchmark is only at two sites in Kentucky and the choice of data sources poses an obstacle to extending the scope elsewhere and outside of the US. The rebuttal explains that while only two US-based regions are included, they are representative of geology elsewhere on the globe, and that there is a roadmap for including more. However, this does not change the state of the current submission. Unresolved.
- Insufficient baselines and evaluation (ASL5, ztAy, QJiJ): Existing multi-modal foundation models that take relevant inputs like DEMs, implicit or coordinate-based models, and non-ML GIS mapping methods deserve comparisons. The rebuttal clarifies the proposed method is intended as a transparent baseline, and not a novel and state-of-the-art architecture, and argues this is consistent with dataset and benchmark conventions (note: this is the inference of the meta-reviewer from reference to "D&B expectations"). Foundation model comparisons are pointed out in the appendix, but only for older models with the exception of Panopticon, and there is a misunderstanding about CLIP models—while the reviewer suggested coordinate-image models like SatCLIP or GeoCLIP the authors understood this to mean image-text models like the original CLIP. The rebuttal also makes a fair point about the difficulty of applying hand-crafted and site-specific features for non-ML GIS methods. While partially addressed, this point is unresolved, because the experimental standard in machine learning for remote sensing papers requires more comparisons (see for example the experiments in Panopticon, Galileo, AnySat, ...).
- Subjectivity in annotations (ztAy): The submission notes that while geological surfaces are continuous, mappers discretize subjectively into classes based on their preferences and expertise. Implications of this subjectivity on training and evaluation are not discussed. The authors state that subjectivity is a genuine characteristic of the domain, and so a strength rather than a weakness of the dataset for its real-world validity. A label-consistency transformation is described in the rebuttal, but there is not a measurement in the dataset of uncertainty or inter-rater variation. While this is a complex aspect of the dataset, and one that may not be possible to address perfectly, this issue is still unresolved in terms of characterizing the effect of this subjectivity as a machine learning benchmark. Potential users will need to be convinced of its validity.
- Insufficient technical novelty in model and analysis (YDrN, ASL5, QJiJ): The proposed SGMap-Net is simple and not novel w.r.t. the topic. The design choices are not ablated and justified in the modeling or the dataset construction. For instance, which modalities are the most important, and does the aggregation method matter? This is partially addressed by clarifications in the rebuttal and in appendix results, but the multiple concerns raised by reviewers indicates that more is needed in the main paper to inform and convince the reader. Unresolved.
- FLOPs and parameter counts (ASL5): These were missing but are included in the revision in Appendix C.2. Resolved.
- Class imbalance could skew evaluation (ASL5, YDrN): There is class imbalance in the annotations, but the authors note this is genuine imbalance in the distribution of the data, and explain the choice to preserve this imbalance and the methodological choices made to mitigate it. Resolved.

**Reviewer Scores:**

Each review is most likely to maintain its score or raise it by one point at most. This prediction is driven by the unresolved and partly addressed issues, even though a rebuttal is provided for each review, because the submission has not substantively changed in a way that would fix the concerns. Ultimately if all scores were raised this would still result in rejection (1, 3) and marginal acceptance (5, 5), and with the unresolved issues these updates would still indicate rejection at this time.

---

### Decision · Program_Chairs · 2026-01-26

Reject